# FlashMD: long-stride, universal prediction of molecular dynamics

**Filippo Bigi**[*]
Institute of Materials
Ecole Polytechnique Fédérale de Lausanne
Lausanne 1015, Switzerland
filippo.bigi@epfl.ch

**Sanggyu Chong**[*]
Institute of Materials
Ecole Polytechnique Fédérale de Lausanne
Lausanne 1015, Switzerland
sanggyu.chong@epfl.ch

**Agustinus Kristiadi**
Department of Computer Science
Western University & Vector Institute
London, ON N6A 3K7, Canada
akristi@uwo.ca

**Michele Ceriotti**
Institute of Materials
Ecole Polytechnique Fédérale de Lausanne
Lausanne 1015, Switzerland
michele.ceriotti@epfl.ch

## Abstract

Molecular dynamics (MD) provides insights into atomic-scale processes by integrating over time the equations that describe the motion of atoms under the action of interatomic forces. Machine learning models have substantially accelerated MD by providing inexpensive predictions of the forces, but they remain constrained to minuscule time integration steps, which are required by the fast time scale of atomic motion. In this work, we propose FlashMD, a method to predict the evolution of positions and momenta over strides that are between one and two orders of magnitude longer than typical MD time steps. We incorporate considerations on the mathematical and physical properties of Hamiltonian dynamics in the architecture, generalize the approach to allow the simulation of any thermodynamic ensemble, and carefully assess the possible failure modes of such a long-stride MD approach. We validate FlashMD's accuracy in reproducing equilibrium and time-dependent properties, using both system-specific and general-purpose models, extending the ability of MD simulation to reach the long time scales needed to model microscopic processes of high scientific and technological relevance.

## 1 Introduction

Simulations of atomic-scale systems are at the core of computational physics, chemistry, biology, and materials science [1]. Molecular dynamics (MD) in particular is a powerful tool for investigating the behavior of microscopic systems, from proteins [2–5] to chemical reactions [6–8] and materials [9–11]. MD elucidates atomic-scale structure and mechanisms by numerically solving the equations of atomic motion, driven by forces that can be estimated by quantum ground state electronic-structure calculations [12]. This has allowed simulating the time-evolution of various systems at the atomic scale, and probing thermodynamic and kinetic properties that are often difficult to measure experimentally.

Despite its utility, MD has long been hindered by the trade-off between computational cost and accuracy of employed methods. Machine learning interatomic potentials [13–19] (MLIPs) have remedied this in part by allowing the cheap approximation of otherwise expensive quantum mechanical

---

[*]These two authors contributed equally to this work.

39th Conference on Neural Information Processing Systems (NeurIPS 2025).

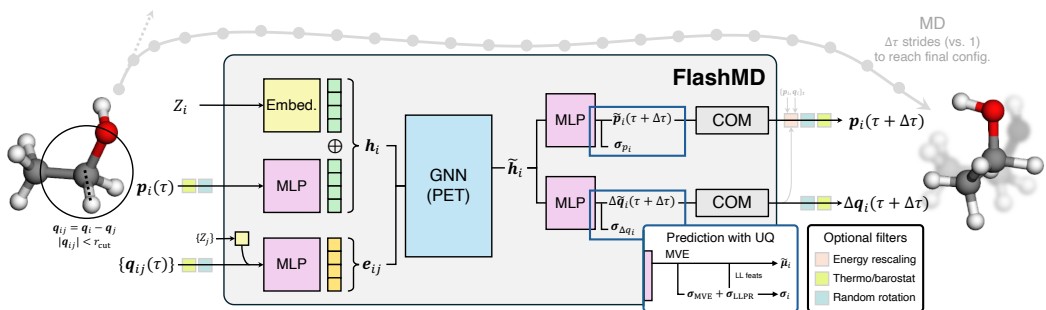

Figure 1: Schematic overview of FlashMD. Atoms of the system at time step $\tau$ are taken as inputs, with atomic numbers $Z_i$ and momenta $p_i(\tau)$ embedded into the node features $h_i$, and relative coordinates $q_{ij}(\tau)$ embedded into the edge features $e_{ij}$ of a GNN for the system. The node outputs are used to predict the new configuration $p_i(\tau + \Delta\tau)$ and $\Delta q_i(\tau + \Delta\tau)$ in a multi-head manner. Center-of-mass constraints are also enforced. Uncertainty quantification can be enabled as shown in the navy inset. Optional filters for energy conservation enforcement, thermodynamic ensemble control, and random rotation are provided, as discussed below. Conventional MD would require $\Delta\tau$ explicit numerical integrations to reach the final configuration as opposed to 1 pass of FlashMD.

energies and forces. Nonetheless, MLIPs exhibit their own constraints of having to obey the physical symmetries and requiring expensive gradient computation to obtain the forces for MD propagation. Furthermore, stable and theoretically meaningful MD simulations require mathematical integration of the equations of motion with sufficiently small time steps ($\sim$ 1 fs), limiting the simulations to a regime far removed from experimentally relevant time scales.

Motivated by the latest developments in MLIPs involving symmetry breaking and direct force learning [20–22], as well as by generative approaches that construct representative atomic configurations without following the physically motivated equations of motion [23], this work will focus on the direct prediction of MD trajectories (see Fig. 1). This approach avoids both the explicit calculation of interatomic forces and the numerical integration of the equations of motion, allowing one to use much larger strides compared to traditional MD integrators – with a corresponding, dramatic extension of the time scales accessible via atomistic modeling.

Our novel contributions are summarized as follows: (i) We provide a thorough theoretical analysis of the problem of directly learning MD trajectories, discussing potential pitfalls. (ii) We introduce techniques for higher accuracy and larger time steps, e.g. by enforcing exact conservation of energy at inference time, proving its importance in stabilizing trajectories and reproducing physically correct behavior. (iii) We generalize the approach to MD in arbitrary, experimentally-relevant thermodynamic ensembles. (iv) We present universal models for direct, large-stride MD, capable of predicting trajectories across a wide range of chemical systems with diverse structure and composition.

## 2   Background and related work

We aim to predict the sequence of position $q_i(\tau)$ and momentum $p_i(\tau)$ for each atom $i$ at the *discrete* time step $\tau$ of a MD trajectory integrated with a small time step $\Delta t$ (so the actual simulation time at a time step $\tau$ is $t = \tau\Delta t$). Conventional MD evolves the dynamics using positions and momenta of all atoms at time step $\tau$ to predict the new positions and momenta at time $\tau + 1$. Our goal here is to be able to take longer strides $\Delta\tau$, skipping all intermediate steps, thereby achieving a corresponding reduction in computational cost and eventually allowing practitioners to access much longer time scales in MD.

### 2.1   Molecular dynamics

In its simplest form, MD is the numerical solution of Hamilton's equations

$$\frac{dq_i}{dt} = \frac{\partial H}{\partial p_i}, \quad \frac{dp_i}{dt} = -\frac{\partial H}{\partial q_i}, \tag{1}$$

for an atomistic system with $N$ atoms at positions $\{\boldsymbol{q}_i\}_{i=1}^N$ and momenta $\{\boldsymbol{p}_i\}_{i=1}^N$. The Hamiltonian function $H$ describing the dynamics, in the absence of external perturbations, takes the form $H(\{\boldsymbol{p}_i, \boldsymbol{q}_i\}_{i=1}^N) = \sum_{i=1}^N \boldsymbol{p}_i^2/2m_i + V(\{\boldsymbol{q}_i\}_{i=1}^N)$, where $m_i$ are the atomic masses and $V(\{\boldsymbol{q}_i\}_{i=1}^N)$ is the potential energy of the system. In practice, Eq. 1 is discretized using a time step $\Delta t$. Among the many algorithms that could be used for numerical integration, the velocity Verlet (VV) algorithm [24] has become the standard due to its simplicity and the fact it preserves exactly some of the key properties of the underlying continuous Hamiltonian dynamics (Sec. 2.2). A single VV step reads

$$\boldsymbol{p}_i \leftarrow \boldsymbol{p}_i - \frac{1}{2}\frac{\partial V}{\partial \boldsymbol{q}_i}\Delta t, \quad \boldsymbol{q}_i \leftarrow \boldsymbol{q}_i + \frac{\boldsymbol{p}_i}{m_i}\Delta t, \quad \boldsymbol{p}_i \leftarrow \boldsymbol{p}_i - \frac{1}{2}\frac{\partial V}{\partial \boldsymbol{q}_i}\Delta t. \tag{2}$$

If integrated with a sufficiently small $\Delta t$, the VV algorithm approximately conserves the energy of the system, making it suitable to sample the $NVE$ thermodynamic ensemble (where the number of particles $N$, volume $V$ and total energy $E$ are fixed).

The $NVE$ ensemble rarely corresponds to realistic experimental conditions. For this reason, variants of MD have been developed to target other types of ensembles [25, 26], accelerate their statistical sampling [27], account for nuclear quantum effects [28], etc. Practically, $NVE$ MD can be modified via the inclusion of thermostats (e.g., [29, 30]), which allows sampling the constant-temperature ($NVT$) ensemble, as well as the addition of barostats to sample constant-pressure ($NpT$) ensemble. Specialized Monte-Carlo moves are also used to access ensembles with varying number of particles and constant chemical potential ($\mu VT$). These different variants of MD are discussed further in Appendix D, and are all built around the $NVE$ integrator, whose accuracy is therefore of central importance to achieve sampling of configurations with the correct probabilities.

**Molecular dynamics with machine learning interatomic potentials** The integrator in (2) is simple and computationally inexpensive, even though it requires a small time step. The bottleneck is typically the evaluation of the force $\boldsymbol{F}_i = -\partial V/\partial \boldsymbol{q}_i$ at every step, which is traditionally done using affordable but inaccurate empirical potentials, or accurate but very demanding quantum mechanical calculations. Over the last two decades, most efforts of applying machine learning (ML) to accelerate MD have revolved around MLIPs that approximate the potential energy surface $V(\{\boldsymbol{q}_i\}_{i=1}^N)$ from quantum mechanical calculations at a much reduced cost. Though MLIPs were traditionally focused on describing a specific chemical system, the last few years have seen the development of "universal" MLIPs [31–40], which aim to provide good accuracy across the whole periodic table, in principle allowing users to simply deploy the model for the desired system without further, dedicated training.

## 2.2 Physical and mathematical considerations

**Symmetries of the potential energy function** The potential $V(\{\boldsymbol{q}_i\}_{i=1}^N)$ obeys two fundamental physical symmetries: (i) $S_N$-invariance: $V(\{\boldsymbol{q}_i\}_{i=1}^N)$ is invariant with respect to permutations of atom indices; (ii) E(3)-invariance: $V(\{\boldsymbol{q}_i\}_{i=1}^N)$ is invariant with respect to translations, rotations and inversions of the atomic structure. Although traditional MLIPs incorporate all these symmetries through symmetry-constrained architectures, some recent models do not directly enforce rotational (and inversion) symmetry and use instead data augmentation strategies to encourage the model to approximately capture it at training-time [18, 22]. This allows the models to avoid expensive equivariant operations and make inference more computationally efficient, without significantly affecting the physical observables in MD [20]. It should also be noted that preserving the two remaining symmetries (those with respect to translations and permutations), which are much harder to augment, naturally leads to the choice of graph neural networks (GNNs) for learning interatomic potentials, explaining their widespread adoption in last-generation MLIPs.

**Conservative and non-conservative forces** Using forces that are the derivatives of $V$ in the propagation of Hamiltonian dynamics conserves the total energy $H$. Even though this is a requirement to sample the $NVE$ ensemble, some recent models have implemented the direct prediction of $\boldsymbol{F}_i$ as an arbitrary vector field, leading to non-conservative dynamics [22, 32, 41]. These "direct-force" models do not even apply data augmentation to encourage energy conservation, as doing so involves computing the Jacobian of the forces, which is computationally impractical. Nevertheless, high model accuracy and the use of hybrid integration strategies [21, 42] allow non-conservative forces to be used in MD without generating noticeable unphysical artifacts, and with the efficiency benefits given by skipping the differentiation step.

**Symmetries in molecular dynamics**    The possibility of performing stable MD with direct force predictions, and therefore without an underlying potential energy surface, suggests that an explicit potential energy function might not be necessary to predict the system evolution over time. It is therefore natural to consider the underlying symmetries in MD, and understand whether they can be incorporated efficiently into a potential-free model, or encouraged at training-time.

The VV algorithm, as presented in Eq. 2, displays two fundamental symmetries, both of which are properties of the underlying continuous solution of Hamilton's equations: (i) MD is *symplectic*. That is, if $\{\boldsymbol{p}_i\}_{i=1}^N$, $\{\boldsymbol{q}_i\}_{i=1}^N$ are the momenta and positions a time step $\tau_1$ and $\{\boldsymbol{p}_i'\}_{i=1}^N$, $\{\boldsymbol{q}_i'\}_{i=1}^N$ are those at a different time step $\tau_2$, then

$$\frac{\partial p_{i,\alpha}'}{\partial p_{i,\alpha}}\frac{\partial q_{i,\alpha}'}{\partial q_{i,\alpha}} - \frac{\partial p_{i,\alpha}'}{\partial q_{i,\alpha}}\frac{\partial q_{i,\alpha}'}{\partial p_{i,\alpha}} = 1, \tag{3}$$

for all $i = 1,...,N$ and $\alpha = x, y, z$. This crucially implies conservation of volume in phase space [43]. (ii) MD is *time-reversible*. This means that, considering positions and momenta evolved from $\tau_1$ to $\tau_2 > \tau_1$, then evolving $\{-\boldsymbol{p}_i'\}_{i=1}^N$, $\{\boldsymbol{q}_i'\}_{i=1}^N$ for $\tau_2 - \tau_1$ time steps will result in the state $\{-\boldsymbol{p}_i\}_{i=1}^N$, $\{\boldsymbol{q}_i\}_{i=1}^N$. Furthermore, (iii) MD is (approximately) *energy-conserving*. While energy conservation is exact for the solution of Eq. 1, its discretization in the form of Eq. 2 is only approximately energy-conserving due to the finite integration step $\Delta t$. Since smaller $\Delta t$ values afford better energy conservation, the latter is an important metric of integrator quality and sampling accuracy, which is why conventional MD is limited to short time steps and therefore simulation times.

**MD as a time series**    The sequence of configurations generated by a MD trajectory obeys a few additional mathematical properties. (i) MD is *Markovian*. It is trivial to see that Eq. 2 defines a Markovian process, i.e. that the present time step $\tau$ contains all the necessary information to predict any future time step $\tau + \Delta\tau$. (ii) MD is *deterministic*. Even though it is possible to describe MD in a probabilistic framework (see e.g. Appendix D), barring effects due to numerical error and parallel programming, the initial conditions determine unequivocally the trajectory, both for infinitesimal and finite time steps. (iii) MD is *chaotic*. Very often, even moderately complex systems simulated with MD exhibit a positive Lyapunov exponent [44] (i.e., the speed at which trajectories diverge), meaning that the phase-space distance of two very close initial states increases exponentially fast as the simulation time progresses. Since MD is executed in finite-precision arithmetic, this implies that the targets of the learning exercise will become excessively noisy (and therefore difficult or impossible to learn) for large strides $\Delta\tau$. Furthermore, the Lyapunov exponent is highly dependent on the physical system under consideration.

## 2.3   ML modeling of molecular dynamics trajectories

A few previous works have applied ML techniques to model MD trajectories or their target distributions. In the following, we provide a non-exhaustive set of related works broadly categorized by their conceptual approach. We focus on methods that aim to avoid VV integration entirely, rather than on methods based on multiple time-stepping [42] that reduce the computational cost by combining the evaluation of cheap-but-inaccurate ML models and more expensive physics [45, 46] or ML-based approximations of $V$ along the trajectory [47]. We also discuss how the nature and the practical implementation of previous approaches compare with the fundamental properties of MD.

**Thermodynamic ensemble generators**    Generative ML techniques have been applied to directly sample the thermodynamically accessible system configurations [48–51], which is especially useful for biomolecular systems with slow conformational transitions. Such models allow cheap prediction of the thermodynamic properties that conventionally require long MD simulations, but they disregard the time-dependent behavior of the systems and hence are unsuitable for investigating the physicochemical phenomena that can only be explained via the system's *dynamics*. More recently, the implicit transfer operator (ITO) has been proposed to extend Boltzmann generators to also recover, with stochastic trajectory, a measure of long-time dynamical processes [52]. Another relevant approach is `Timewarp` [53], which employs a normalizing flow as a proposal distribution within a Markov chain Monte Carlo scheme targeting the Boltzmann distribution, achieving effective time steps on the order of $10^5$–$10^6$ fs for molecular systems.

**Time-series approaches**    Several works [54–57] have interpreted the MD trajectory as a time series and adopted recurrent neural network (RNN)-type architectures, particularly the long short-

term memory [58] (LSTM), for the learning task. These models take a time series of past system configurations as inputs to make predictions of the future trajectory. Some have taken a stochastic approach to predict a probable *distribution* of system states at a future time, and this has been successfully demonstrated in both all-atomic [54] and coarse-grained [56] contexts. However, in light of the Markovian nature of MD, sequence models such as LSTM use redundant information and are not necessary to learn MD trajectories. Furthermore, the deterministic nature of MD as presented in Eq. 2 makes it superfluous to use probabilistic models (VAEs [59], normalizing flows [60], diffusion models [61], etc.).

**Direct MD propagators** This is the class of methods that most closely resembles our approach. It is distinct from the former two approaches in that no generative approaches or multiple time step information is used. In this case, the ML models take only the positions and momenta of atoms of the system at time step $\tau$ as inputs and deterministically predict their changes at $\tau + \Delta\tau$, hence "directly propagating" the dynamics. In MDnet by Zheng et al. [62], the chemical system is described as a graph, with the edge features incorporating both the relative positions and momenta between the atoms. The model then predicts the changes in the positions and momenta for a fixed large time step $\Delta t$. Very recently, Thiemann et al. have demonstrated TrajCast [63], an autoregressive equivariant network for direct MD prediction. Their framework has been shown to achieve good accuracy in reproducing $NVT$ trajectories for individual molecular or bulk systems at a specific thermodynamic state point. It is also worth mentioning GICnet [64] and its transferable, transformer-based variant MDtrajNet-1 [65], which learns a function that takes as inputs the initial positions, velocity, and $\Delta t$ to return the positions at time $t + \Delta t$, the Equivariant Graph Neural Operator [66] (EGNO) approach, which predicts the evolution of the system at multiple times using equivariant temporal convolution in Fourier space, and the Graph Network-based Simulators [67, 68] (GNS), which have been developed for arbitrary particle-based systems without the chemical context.

## 3 The FlashMD framework

In this section, we explain the design choices made for "FlashMD", our proposed approach for the direct learning and prediction of MD trajectories.

### 3.1 Learning molecular dynamics trajectories with graph neural networks

Given that MD shares with interatomic potentials the properties of $E(3)$-equivariance and the use of atomic geometries as inputs, we propose that FlashMD should be built on top of similar GNN architectures to those that have successfully been used to model machine-learning interatomic potentials (MLIPs). In this work, we choose the Point-Edge Transformer [18] (PET), although any GNN architecture could be used. Compared to the original architecture in Ref. [18], we make two physically motivated modifications to adapt it to the prediction of trajectories: (i) Each node state is also initialized using the particle momentum $\boldsymbol{p}_i$, encoded via a multi-layer perceptron, in addition to the chemical species of the atom under consideration. The initialization of the edge states remains unchanged, and it includes the interatomic vectors $\boldsymbol{q}_j - \boldsymbol{q}_i$. (ii) The outputs $\boldsymbol{p}_i(\tau + \Delta\tau)$ and $\boldsymbol{q}_i(\tau + \Delta\tau)$ are node properties and are therefore predicted via two distinct multi-layer perceptron heads starting from the node representation of PET.

It should be noted that the "raw" FlashMD predictions are chosen to be mass-scaled, i.e. $\boldsymbol{p}_i'/\sqrt{m_i}$ and $(\boldsymbol{q}_i' - \boldsymbol{q}_i)\sqrt{m_i}$. This ensures a treatment of displacements and momenta on equal footing for atoms of different mass, although a data-driven approach is also possible (App. A). Further details on the architecture and the training procedure are available in Appendices A and B, respectively.

### 3.2 Addressing the many pitfalls of direct molecular dynamics predictions

Despite the fact that we have identified and justified graph neural networks as a highly-suitable model to predict MD trajectories, there still exist many problematic aspects of this exercise which, if ignored, could make the resulting models practically useless. We note that these, as well as many of the theoretical considerations above, have been almost entirely ignored in previous related works.

**Out-of-distribution predictions** Robust *epistemic* uncertainty schemes, capable of predicting errors associated with limited data sampling, are generally highly recommended when sampling

configurations using MLIPs [69]. They become essential for models that predict MD trajectories directly, that are less physically grounded and more susceptible to exhibiting pathological and unphysical behavior when queried outside of the domain of their training data.

**Chaoticity**   The chaoticity of MD limits the time scale that can be reached with deterministic predictions. It also introduces an *aleatoric* component to the model error, which varies in intensity depending on the system (see Sec. 2.2) and should be accounted for when building an uncertainty quantification scheme. For more information, see App. M.

**Time-reversibility**   Time-reversibility, one consequence of the symmetries of MD, can easily be incorporated by data augmentation [62]. This follows trivially from the discussion in Sec. 2.2.

**Conservation of energy**   Conservation of energy is another consequence of the fundamental symmetries of MD, namely the translational symmetry of time. Given the radical importance of translational symmetries in 3D space, which make GNNs so effective for MLIPs, it is clear that conservation of energy should be considered a centerpiece of MD trajectory modeling, as it encodes the symmetry in the time dimension. Previous works have not carefully monitored conservation of energy in their predicted MD trajectories. In FlashMD, we implement two approaches to improve energy conservation: (i) we utilize errors in energy conservation during training, in addition to the terms corresponding to the position and momenta (see App. B), (ii) we enforce energy conservation at inference time by rescaling momenta after each FlashMD step (see App. C). The latter adjustment makes it possible to run long trajectories targeting the $NVE$ ensemble, as these would otherwise be affected by a large energy drift (see App. F).

**Symplecticity**   Symplectic behavior is – together with energy conservation – a necessary and sufficient condition for correct thermodynamic sampling. Unfortunately, penalizing non-symplecticity in the loss function is impractical, as evaluating Eq. (3) involves the computation of the full $3N \times 3N$ Jacobian – similar to energy conservation in direct force prediction [21]. We will discuss some numerical results on the violation of (3) by FlashMD, but mainly focus on the empirical measures of the accuracy of dynamics and sampling, through comparison with conventional MD simulations.

**Symmetry breaking**   Although equivariant GNNs include strict rotational symmetries in the model, many GNN architectures do not enforce rotational equivariance explicitly [18, 36, 38]. Given that directly learning MD trajectories is a fundamentally more challenging problem than learning a potential energy surface, symmetry breaking might affect models for MD more than MLIPs. Therefore, if using rotationally unrestricted GNNs, we recommend correcting for rotational (and inversion) symmetry breaking at inference time, using similar techniques as those proposed for MLIPs [20]. Given that the PET architecture [18] also does not enforce rotational equivariance, we use rotational and inversion augmentation at training time, and optionally perform random rotation(s) of the system at each step of FlashMD simulations (see App. A).

### 3.3   Generalization to arbitrary thermodynamic ensembles

FlashMD is trained to reproduce, with a longer stride, $NVE$ trajectories obtained with a VV integrator. However, nearly all other MD variants can be discretized (and are often implemented) using a split-operator formalism where VV is one of the components of the algorithm for a single time-step. This construction, which is further discussed in App. D, allows using FlashMD to accelerate the majority of MD variants and ensembles.

## 4   Results

To demonstrate the capabilities of FlashMD, we trained two types of models: *water-specific* models trained on a dataset of MD trajectories for liquid water, and general-purpose, *universal* models trained on MD trajectories of structures sampled from the MAD dataset (see Ref. [39]). All reference MD simulations were performed with the PET-MAD universal MLIP [39]. For both the water-specific and universal cases, we trained separate models targeting different time strides. Full training details are available in App. B. The following subsections will focus on the testing of such models in predicting meaningful physical observables for the corresponding systems. Ablation studies are discussed in

App. F, and timings are provided in App. G. Test set accuracy benchmarks against MDNet [62] and TrajCast [63] are reported in App. H and App. I, respectively.

## 4.1 Liquid water

Liquid water is central to many physical, chemical, biological and environmental processes, in great part thanks to its microscopic hydrogen-bonding network and resulting physical and chemical properties. The study of liquid water at the microscopic level with MD is therefore a very active area of research [70–73], and we use it here as an example of how FlashMD models can accurately predict physical observables at the molecular level. For consistency with the universal model, we train the water-specific model on trajectories obtained with PET-MAD, even though its reference electronic-structure method (PBEsol [74]) is known to grossly overestimate the melting point of water. For this reason, we perform simulations with a target temperature of 450 K, above the melting temperature of this model. Results for the q-TIP4P/f model [75] at room temperature are discussed in Appendix J.

We focus on the evaluation of static observables, i.e. the equilibrium, time-independent properties of a physical system which can be estimated as averages over MD trajectories. As easy-to-compute diagnostics, we investigate the mean kinetic energy and atomic radial distribution functions in $NVT$ simulations, as well as the equilibrium density predicted by $NpT$ simulations. The mean kinetic energies (expressed as effective temperatures) are shown in Table 1. It can be seen that, while the models without energy conservation can show pathological deviations from the target temperatures, the energy conservation enforcement approach described in App. C always recovers the correct temperatures for the overall simulation. However, significant deviations can still be observed for the global stochastic velocity rescaling [30] (SVR) thermostat in the kinetic energies resolved by atom type. This is a spurious effect (classically, each degree of freedom should have a mean kinetic energy equal to $k_B T/2$) which is also observed in non-conservative force models [21]. The link to direct force prediction (that can be seen as the $\Delta t \to 0$ of trajectory prediction) is also suggested by the fact that lack of equipartition is observed also for short-time FlashMD models, that have excellent validation accuracy. As a consequence, one needs to employ *local* thermostats, such as those based on Langevin dynamics, similar to what was done in Bigi et al. [21]. With a judicious choice, one can achieve accurate sampling of equilibrium properties, without reducing substantially sampling efficiency. However, local thermostats disrupt dynamical properties, to an extent that depends on the strength of the thermostat coupling, and a thorough quantitative analysis of dynamical properties require disentangling the effect of the long-stride sampling and that of the thermostats needed to obtain accurate equilibrium sampling (see App. J). For these reasons, we primarily focus our quantitative analysis on time-independent equilibrium properties, and discuss examples where FlashMD qualitatively captures time-dependent behavior.

The atomic radial distribution functions (for the oxygen and hydrogen atoms, respectively), are shown in the left and center panels of Fig. 2. Here, it can be seen that FlashMD is able to correctly reproduce the distribution functions from the reference MD simulations, for both water-specific and universal models. To demonstrate the behavior when simulating the constant-pressure $NpT$ ensemble, we also compute densities at ambient pressure (Fig. 2, right panel). The water-specific FlashMD models are

Table 1: Difference between effective and target temperatures in $NVT$ simulations of liquid water using FlashMD, using different models and thermostats, and comparing results with and without enforcement of energy conservation. Characteristic times of 100 fs and 10 fs were used for the Langevin and SVR thermostats, respectively. The "all", "O", "H" labels refer to the subset of atoms under consideration. Subscripts on the results refer to statistical sampling errors. All units are in Kelvin; numbers close to zero are better.

| Model | Without energy conservation | | | | | | With energy conservation | | | | | |
| | Langevin | | | SVR | | | Langevin | | | SVR | | |
| | $\Delta T_{\text{all}}$ | $\Delta T_{\text{O}}$ | $\Delta T_{\text{H}}$ | $\Delta T_{\text{all}}$ | $\Delta T_{\text{O}}$ | $\Delta T_{\text{H}}$ | $\Delta T_{\text{all}}$ | $\Delta T_{\text{O}}$ | $\Delta T_{\text{H}}$ | $\Delta T_{\text{all}}$ | $\Delta T_{\text{O}}$ | $\Delta T_{\text{H}}$ |
|---|---|---|---|---|---|---|---|---|---|---|---|---|
| Water, 1 fs | -1.3(0.9) | -1.1(1.3) | -1.4(0.9) | -0.4(0.3) | 13.8(7.0) | -7.4(2.3) | -0.3(0.8) | -1.6(1.4) | 0.3(0.9) | -0.3(0.3) | -5.4(4.3) | 2.2(2.2) |
| Water, 4 fs | 1.4(0.9) | -1.4(1.2) | 2.8(1.2) | 0.1(0.3) | -21.4(2.9) | 10.8(1.4) | -0.4(0.9) | -3.6(1.3) | 1.2(1.1) | -0.1(0.3) | -20.4(2.7) | 10.0(1.1) |
| Water, 16 fs | -0.2(0.8) | -1.1(1.2) | 0.2(0.8) | -0.1(0.4) | -16.0(2.9) | 7.8(1.3) | 1.3(0.9) | 1.2(1.0) | 1.4(1.1) | 0.1(0.4) | -10.7(2.1) | 5.4(1.2) |
| Universal, 1 fs | 33.8(1.0) | 36.2(1.9) | 32.8(1.1) | 8.0(0.4) | -57.5(5.5) | 40.4(1.9) | 0.2(0.8) | -1.4(1.1) | 0.9(1.0) | -0.5(0.4) | -58.6(2.8) | 28.2(1.6) |
| Universal, 4 fs | 10.7(1.0) | 7.3(1.3) | 12.5(1.3) | 9.2(1.4) | 79.2(3.1) | -25.3(2.2) | -0.7(0.8) | -1.9(1.4) | -0.1(1.0) | 0.4(0.4) | -31.2(4.6) | 16.0(2.0) |
| Universal, 16 fs | -22.5(0.7) | -20.9(1.1) | -23.5(0.9) | -4.0(0.4) | 7.6(2.3) | -9.8(1.4) | 0.4(0.9) | 2.9(1.3) | 0.8(1.0) | 0.1(0.4) | 8.6(3.0) | -4.1(1.4) |

able to reproduce densities similar (although not statistically equivalent) to the reference MD. The universal models show significant deviations from the reference calculation, although smaller than the typical discrepancy expected when varying the details of the electronic structure calculation.

## 4.2 Universal long-stride sampling

Having compared and validated both system-specific and universal FlashMD models on the liquid water system, we now proceed to demonstrate the accuracy of the universal FlashMD models for more complex and chemically-diverse systems. We consider three archetypal examples that showcase the relevance for different classes of chemical and materials science problems: (i) We estimate the distribution of Ramachandran angles for alanine dipeptide, a system that exhibits the basic features of protein dynamics and that is often used to benchmark sampling methodologies in biomolecular simulations; (ii) We model the finite-temperature dynamics of the Al(110) surface, a deceptively simple system that exhibits spontaneous formation of defects and surface pre-melting [76]; (iii) We compute the temperature-dependent diffusion coefficient of Li atoms in lithium thiophosphate (LiPS) a material that is being investigated as an electrolyte for solid-state batteries [77, 78], and that allows us to demonstrate the accuracy in capturing time-dependent properties.

**Solvated alanine dipeptide** We extend upon the previous liquid water simulations, generating constant-pressure trajectories of a single alanine dipetide molecule in solution, following closely the setup described in Morrone et al. [79]. The energy landscape is probed in three different simulations: MD using the PET-MAD potential with 0.5 fs stride, and universal FlashMD with 8 fs and 16 fs strides. Note that for this system 0.5 fs is a limit above which MD simulations exhibit severe instabilities. The Ramachandran plots (Figure 3a) show the characteristic distribution of molecular conformers in terms of the backbone dihedral angles. This is a model for the backbone flexibility of proteins and demonstrates the ability of FlashMD to recover major features of the Ramachandran plot (particularly the low-energy basins in the $-\pi \leq \phi \leq 0$ range) with strides up to 32 times larger.

**Metal surface** As discussed in a previous work [76], the premelting behavior in Al(110) is characterized by two soft vibrational modes of the surface atoms: the top layer atoms show softening in the [001], or $x$ component, and the second layer atoms show softening in the [110], or $z$ component. Figure 3b shows that the universal Flash MD models correctly describe this trend, with the mean square displacement (MSD) larger for the corresponding surface atoms and their associated softer vibrational modes at 500 K. The dynamical consequences of premelting is presented in the trajectory traces of 3b, which is generated from the FlashMD simulation with 64 fs strides at 600 K. The trajectory traces provide a visual representation of the anisotropic softening of vibration for first and second layer atoms, as well as the dynamic adatom formation pathways involving cooperative migration of both the surface and second layer atoms and effective adatom diffusion via exchange with nearby surface atoms. This demonstrates the ability of FlashMD to not only capture material specific trends, but also describe meaningful dynamical behavior, despite the much larger strides.

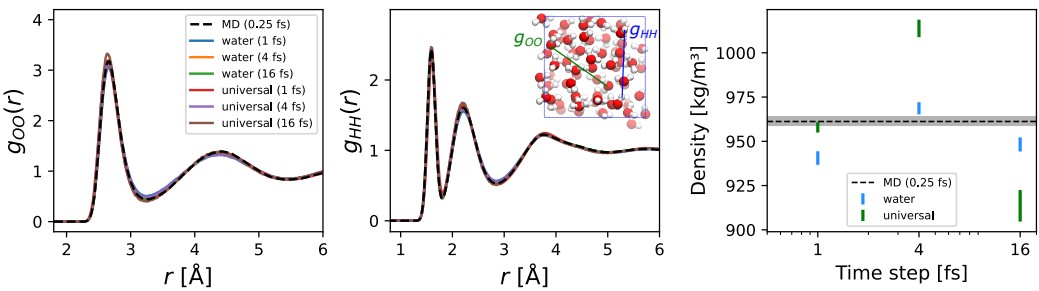

Figure 2: Comparison of physical observables obtained from MD (black) and FlashMD (other colors). Left and Center: radial distribution functions for oxygen and hydrogen atoms, respectively, from simulations in the $NVT$ ensemble using the Langevin thermostat. Right: densities from simulations in the $NpT$ ensemble.

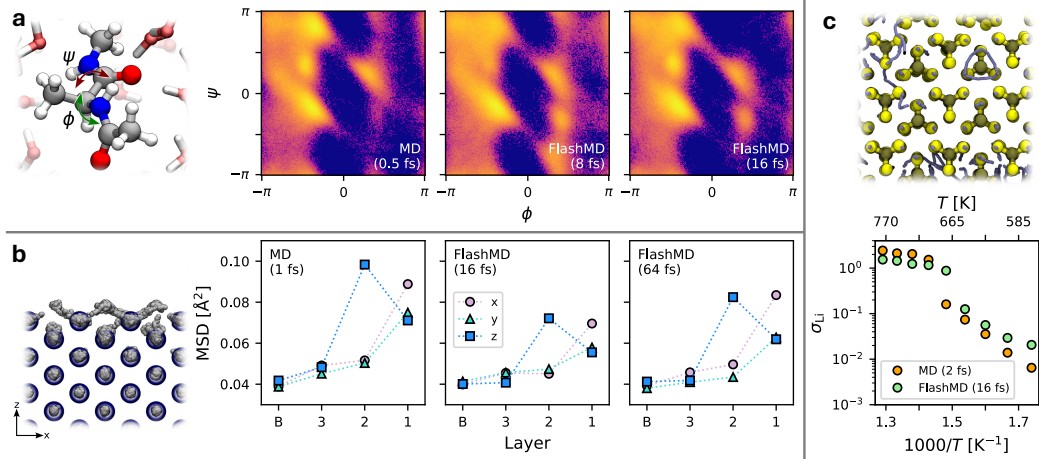

Figure 3: Results of case studies conducted for the universal FlashMD models. (a) Ramachandran plots of the main backbone dihedrals for a simulation of solvated alanine dipeptide at 450 K. (b) Mean square displacement (MSD) of the Al (110) surface atoms at 500 K, at different layers from the surface (B indicates the limiting value for the bulk). The premelting and defect formation phenomena are also visualized as traces of atomic positions from a FlashMD simulation at 600 K, run with $\Delta\tau = 64$ fs. The ideal atomic positions are also shown for reference. (c) Li conductivities of $\gamma-Li_3PS_4$ at varying $T$, along with the initial system configuration overlaid with traces of the Li atom positions from the FlashMD trajectory at 700 K. In both (b) and (c), traces are obtained with a moving average in time to remove thermal fluctuations and visualize more clearly the diffusive behavior.

**Solid-state electrolyte**   At high temperatures, the $\gamma$ phase of lithium thiophosphate undergoes a phase transition to a superionic state that exhibits much higher conductivity. We reproduce simulations analogous to those in Ref. [80], computing the Nernst-Einstein conductivity of a LiPS cell as a function of temperature. Results in Figure 3c show that the universal FlashMD model successfully describes the superionic transition of $\gamma$ Li$_3$PS$_4$ and predicts for it to take place at 675 K, within the established transition temperature range determined in previous simulations using PET-MAD [39]. Li conductivities are reasonably matched with the reference MD simulations, albeit with systematic over- and under-estimations in the low and high $T$ regimes, respectively.

## 5   Discussion

Machine learning has been quietly revolutionizing the atomistic modeling of matter, accelerating the most time-consuming parts of physics-based calculations while striving to retain as much as possible of the underlying physical symmetries and constraints. As datasets and models grow in scale, there is increasing interest in more radical approaches that trade the physical grounding of established practices for computational efficiency. Our work demonstrates that there is enormous potential in constructing GNN models that predict directly the evolution of atomic coordinates and momenta, allowing MD simulations to propagate with long strides, each replacing tens of costly force evaluations with miniscule time steps. Contrary to the very few previous works in this direction, which were restricted to reproducing MD trajectories for a specific system in prescribed thermodynamic conditions, we show that our FlashMD architecture allows one to obtain a *universal* direct MD model that can be applicable to different thermodynamic conditions and ensembles, and to wildly diverse atomic structures and compositions.

This is not to say that circumventing Hamiltonian dynamics is without problems. We highlight several ways an architecture similar to FlashMD could fail, by breaking some of the fundamental symmetries and conservation laws that are obeyed (at least approximately) by conventional integrators. We show how these shortcomings can be mitigated, e.g., by performing energy rescaling at inference time, or by including thermostats to control systematic drifts in the constant-energy trajectories. While the design choices of FlashMD deviate from Hamiltonian dynamics, an alternative framework that preserves symplecticity (and therefore Hamiltonian dynamics) in the direct learning of trajectories is

proposed elsewhere [81]. Many of the shortcomings of non-symplectic FlashMD are shared by non-conservative force models, which have also become fashionable as a tool to accelerate MD. We argue that the transformative speed-up afforded by FlashMD makes direct MD trajectory prediction a more promising approach, at least when performing exploratory studies that require simulating long time scales. As shown concretely by challenging examples that simulate with semi-quantitative accuracy three archetypal systems for biochemistry, surface science and energy technologies, FlashMD can already be applied to realistic simulation problems, capturing the essential equilibrium and dynamical processes while accelerating sampling significantly in all cases (App. G).

In considering potential directions for further development, one should keep in mind that, contrary to the case of ML interatomic potentials that have been studied in great detail and brought to scale over the past decade, there is very little existing research on direct MD prediction. We recognize the possibility of incorporating further constraints in the model architecture, or refinements to the training details, that can better enforce the conservation laws obeyed by the fine-grained VV integrator. One could also investigate scaling up the FlashMD universal model to more parameters and a larger trajectory datasets, or implement a modified architecture that targets multiple time strides with a single model. Given that training relies on short MD trajectories built from a universal MLIP, it is relatively affordable to increase the dataset size by at least one order of magnitude. Moving forward, we envisage a future in which every MLIP would come with its own FlashMD-style long-stride MD companion models, increasing even further the time scales within reach of ML-driven atomic-scale simulations.

## Software and data

The FlashMD models presented in this work were trained with the `metatrain` package [82], and they support inference in multiple simulation engines (including ASE [83], i-PI [84], LAMMPS [85]) through `metatomic` [82].

Helper functions to download universal FlashMD models and to prepare simulations are distributed with the `flashmd` package available on PyPI. Further information and instructions can be found at https://flashmd.org, including links to the training datasets and scripts to reproduce the reported results on HuggingFace and Materials Cloud [86]. An example of the use of FlashMD is also available at https://atomistic-cookbook.org/examples/flashmd/flashmd-demo.html.

Besides the universal models presented in this work, which were trained to reproduce molecular dynamics at the PBEsol level of theory (through the PET-MAD universal potential), we also make available FlashMD models based on the more accurate $r^2$SCAN functional which we recommend for scientific use. A separate set of case study results for these universal FlashMD models are presented in App. L.

## Acknowledgments and Disclosure of Funding

The authors would like to thank Davide Tisi and Federico Grasselli for help in processing LiPS trajectories. FB and MC acknowledge support from the NCCR MARVEL, funded by the Swiss National Science Foundation (SNSF, grant number 182892) and from the Swiss Platform for Advanced Scientific Computing (PASC). MC and SC acknowledge funding from the Swiss National Science Foundation (Project 200020_214879).

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

# A  Model details

## A.1  Why predict $q_i' - q_i$ and $p_i'$: small- and large-time limits

It should be noted that the models for liquid argon in the early work by Zheng et al. [62] effectively predict $q_i' - q_i - p_i \Delta t / m_i$ and $p_i' - p_i$. While the additional terms with respect to FlashMD ($-p_i \Delta t / m_i$ and $-p_i$, respectively) reduce the variance of the targets for small time steps and/or very smooth potential energy surfaces (such as Lennard-Jones argon in Ref. [62]), they instead increase it for more complex systems and larger predicted time steps, which are the focus of the present work. This is a consequence of more complex systems generally having smaller position and momentum correlation times. As a result, we do not shift the targets by these additional terms in our work.

## A.2  Predicting mass-scaled positions and momenta

All models shown in this work are trained on, and therefore predict, mass-scaled displacements and momenta defined as $\Delta \tilde{q}_i = \Delta q_i \sqrt{m_i}$ and $\tilde{p}_i = p_i / \sqrt{m_i}$, respectively, where $m_i$ is the mass of atom $i$. This is aimed at making the scales of the training targets uniform across atoms of potentially very different mass, and it is of fundamental importance for models trained on the whole periodic table. This prevents, for example, the displacement of light atoms or the momenta of heavy atoms from dominating the loss, instead leading to good predictions for all atoms, independent of their mass. At prediction time, a simple scaling using the masses recovers the conventional displacement and momenta. We found that a similar, but data-driven, approach can provide the same benefits. This consists of using different standardization factors for different chemical elements, so that training displacements and momenta are scaled to unit standard deviation during training, *for each chemical element* (i.e., two scaling factors are used for every single chemical element in the dataset: one for displacements, one for momenta).

## A.3  Center-of-mass enforcement

Using Eqs. 2 to evolve a system without external forces naturally leads to conservation of total momentum of the system, i.e.,

$$\sum_{i=1}^{N} p_i' - \sum_{i=1}^{N} p_i = \mathbf{0}. \tag{4}$$

Since the total momentum is constant, the center of mass of the system follows a uniform linear motion, i.e.,

$$\sum_{i=1}^{N} m_i q_i' - \sum_{i=1}^{N} m_i q_i = \Delta t \sum_{i=1}^{N} p_i. \tag{5}$$

Both conditions are enforced within the model to avoid unphysical drift effects during molecular dynamics simulations (Fig. 1). We note that removing the center-of-mass motion entirely would not be correct in the general case, although many MD simulations are performed with this additional constraint. Although we enforced these contraints within the model in this work, we recommend applying them at inference time only in order not to break the locality assumption that underpins our approach.

## A.4  Optional inference-time filters

Within FlashMD, we have implemented "filters" (Fig. 1) that can be employed at inference-time to mitigate the artifacts of direct MD prediction. We refer to the dedicated sections for energy conservation enforcement (App. C) and thermodynamic ensemble control (App. D). Here we only provide a discussion of the random rotation filter.

Since equivariance is not exactly preserved and only learned via data augmentation in the case of unrestricted architectures such as PET [18], simulations performed with the resulting FlashMD model would be prone to spurious effects. To mitigate this, we adopt the strategy proposed Langer et al. [20], where random rotations of the simulated system are performed to average out any artifacts of non-equivariance along the MD trajectory. Implementation details of the random rotation filter is shown in Fig. 4. We note that in case of GNNs that preserve rotational symmetry, this filter is not needed – and would actually have no effect if applied.

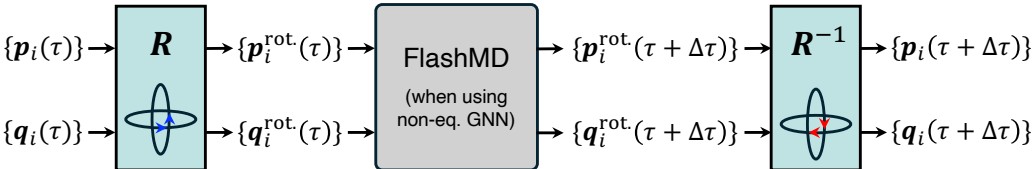

Figure 4: Implementation of the random rotation filter. A random rotation matrix $\boldsymbol{R}$ is sampled and applied on all coordinates and momenta before the rotated inputs are provided to FlashMD. After model inference, $\boldsymbol{R}^{-1}$ is applied to rotate the system back to the original coordinate reference. Random rotation filter is only relevant for rotationally unconstrained GNNs.

# B  Training details

## B.1  Loss function, optimization, normalization

All models are implemented in PyTorch [87] and, unless specified otherwise (see App. E), are trained using a loss function given by the sum of two mean square error terms, for the mass-scaled momenta and the positions respectively:

$$\mathcal{L} = \sum_{s=1}^{N_{\text{train}}} \frac{1}{3N_s} \sum_{i=1}^{N_s} (\tilde{\boldsymbol{p}}_i' - \tilde{\boldsymbol{p}}_{i,\text{ref}}')^2 + (\Delta\tilde{\boldsymbol{q}}_i' - \Delta\tilde{\boldsymbol{q}}_{i,\text{ref}}')^2, \tag{6}$$

where $s$ is an index for structures in the training set and $N_s$ is the number of atoms in structure $s$. In order to ensure similar weight in the loss function between position and momentum terms, the mass-scaled positions and momenta are scaled by their standard deviation across the dataset before training.

Optimization is carried out using the Adam [88] optimizer with an initial learning rate of $3 \cdot 10^{-4}$. Learning rate decay is applied at a regular intervals of 100 and 50 epochs for the water and universal models, respectively. Training-time rotational augmentation for vectorial targets is carried out in the same way as in Pozdnyakov and Ceriotti [18].

## B.2  Training-time energy conservation

We found that the degree of energy conservation on structures of the validation set correlates well with the quality of the models during molecular dynamics runs. As a result, during training, we choose the best model as the one having the lowest product of three terms, evaluated across the validation set: (i) RMSE on the predictions of $\tilde{\boldsymbol{p}}'$, (ii) RMSE on the predictions of $\tilde{\boldsymbol{q}}'$, and (iii) RMSE on the energy of the resulting structure when compared to the energy of the target structure. While an energy term might also be included in the loss function, we found that it slows down training significantly (due to evaluations of the energy model and its gradients), without improving the quality of the FlashMD models in any measurable way.

Indeed, as shown in App. F, models with similar accuracy on positions and momenta can predict MD states with highly varying degrees of energy conservation. In particular, energy misalignment is often observed if the error on the energies is ignored. Although we found this approach to improve the quality of the simulations afforded by our water models (both PET-MAD and q-TIP4P/f), we found its impact on universal models to be less dramatic.

We also found it useful to compare errors in total energies with familiar metrics such as "chemical accuracy" or the accuracy of the underlying energy model. For all models tested in this work, such comparisons correlate extremely well with the quality of the models in the resulting MD simulations.

## B.3  Reference MD trajectory generation

All reference MD trajectories were obtained from simulations performed with PET-MAD [39] (version 1.0), a universal MLIP capable of making reasonably accurate predictions of the potential energy surface across the entire periodic table of elements. All simulations were performed using the Atomic Simulation Environment [89] (ASE) software (version 3.24).

**Water-specific models** A water structure at experimental density (at 298 K and 1 atm) was equilibrated with PET-MAD (or q-TIP4P/f for the q-TIP4P/f-based water models discussed in Appendix J). Subsequently, two more structures were generated by increasing and decreasing the volume of the cell by 10%, scaling the atomic positions accordingly. For each of the three resulting structures, $NVT$ equilibration runs were performed at all temperatures between 20 and 1000 K, in steps of 20 K, with a time step of 0.5 fs and a duration of 5 ps, using a Langevin thermostat with a characteristic time of 10 fs. Subsequently, each equilibrated structure was used to produce an $NVE$ MD trajectory of 2 ps with a time step of 0.25 fs. Structures for training were extracted from these trajectories every 100 fs, and augmented with their time-reversed version, for a total of 5400 structures.

**Universal models** 10,000 structures from the MAD dataset, used in the training of PET-MAD [39], baseline MLIP, were randomly selected for reference MD trajectory generation (see Ref. [39] for further details on the MAD dataset). The initial geometry was first energetically optimized with the BFGS algorithm until the maximum force component threshold of 0.01 eV/Å was reached. The energy-optimized system was put through equilibration under the $NVT$ ensemble for 10 ps with timesteps of 0.5 fs. A characteristic time of 100 fs was used in the Langevin thermostat. The final configuration from $NVT$ equilibration was then taken for trajectory production under the $NVE$ ensemble for 2.5 ps with finer timesteps of 0.25 fs. Positions and momenta were recorded every timestep for FlashMD training. Simulations were repeated 10 times for each structure, with a randomly selected temperature between 0 and 1500 K. Structures for training were chosen from these trajectories every 500 fs (5 samples per $NVE$ trajectory to avoid time-correlated samples), and augmented with their time-reversed version, for a total of 1 million structures.

## C   Enforcing conservation of energy by momentum rescaling

Especially for $NVE$ simulations, it is important to avoid excessive energy drift. In practice, we found that enforcing total energy conservation after each step is beneficial (even for thermostatted runs, see Sec. 4). This is achieved by rescaling the momenta in the following way:

$$\boldsymbol{p}'_i \leftarrow \alpha \boldsymbol{p}'_i, \quad \alpha = \sqrt{1 - \frac{E' - E}{K'}}, \tag{7}$$

where $E'$ is the total energy after the step, $E$ is the total energy before the step, and $K'$ is the kinetic energy after the step.

It should be noted, however, that enforcing energy conservation requires one or two energy evaluations per step (depending on the ensemble and integration scheme), potentially introducing significant overhead. Implementation details of the energy conservation enforcement filter in FlashMD is visualized in Fig. 5.

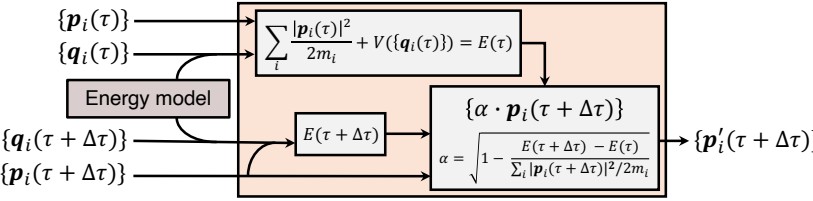

Figure 5: Implementation details of the energy conservation enforcement filter in FlashMD. Energy model can be any model of the interatomic potential (e.g. MLIP, classical force field, etc.) that can be used to compute $V(\{\boldsymbol{q}_i\})$.

## D   Arbitrary thermodynamic ensembles with FlashMD

Molecular dynamics trajectories conserve, at least approximately, the classical total energy of the system, which makes it appropriate for sampling configurations under constant energy, volume and particle number ($NVE$) conditions. Several modifications to Hamiltonian dynamics have been proposed [25, 26] to generate configurations consistent with other thermodynamic conditions, such as constant temperature ($NVT$), constant pressure ($NpT$), or constant chemical potential ($\mu VT$). These ensembles are often more relevant to compare with experimental conditions, that usually do not involve closed, isolated setups – especially not on the small length scales that are used in simulations. Here we discuss two approaches that are routinely used to this end in MD, and how they can be can be combined with FlashMD to extend its constant-energy long-stride integration to sample other ensembles.

First, given that $NVE$ trajectories conserve not only the total energy, but also the probability measures associated with most other ensembles, it is possible to alternate segments of $NVE$ trajectories with discrete Monte Carlo moves changing the particle velocities, the simulation cell size, or the nature of the atoms, using a Metropolis-Hastings criterion [90] to accept or reject them in a way that is consistent with the desired ensemble. This approach can be applied straightforwardly to FlashMD, and its reliability depends on the assumption that segments of FlashMD trajectories are symplectic and energy-preserving to a high degree of accuracy, which is why we give much emphasis to these diagnostics in our study.

The second approach is slightly more subtle and requires some additional technical background, and we discuss and test it in more detail. Integrators for Hamiltonian dynamics can be expressed in a Liouvillian formalism, in which the trajectory density $P(\boldsymbol{q}, \boldsymbol{p})$ is evolved in time according to an operator that combines the time evolution of the different variables, e.g. for Hamiltonian dynamics

$$\mathrm{i}\hat{L} = \sum_i \frac{\partial H}{\partial \boldsymbol{p}_i} \cdot \frac{\partial \square}{\partial \boldsymbol{q}_i} - \frac{\partial H}{\partial \boldsymbol{q}_i} \cdot \frac{\partial \square}{\partial \boldsymbol{p}_i} = \mathrm{i}\hat{L}_q + \mathrm{i}\hat{L}_p \tag{8}$$

Finite-time propagation of the trajectory density can be formally achieved with an exponential operator $e^{\mathrm{i}(\hat{L}_q + \hat{L}_p)\Delta t}$, and the difficulty in developing an exact propagation algorithm for $(\boldsymbol{q}, \boldsymbol{p})$ can be understood as a consequence of the fact that $\hat{L}_q$ and $\hat{L}_p$ do not commute. The error grows with the time step $\Delta t$, and can be reduced using symmetric Trotter factorizations such as $e^{\mathrm{i}\hat{L}_p \Delta t/2} e^{\mathrm{i}\hat{L}_q \Delta t} e^{\mathrm{i}\hat{L}_q \Delta t/2}$. This splitting corresponds, in the trajectory picture, to the VV integrator in Eq. (2). Continuous equations of motion that describe other thermodynamic ensembles can be derived with an extended Lagrangian formalism, in which additional structural parameters (e.g. the simulation cell volume $V$ or shape) are associated with fictitious masses and momenta. Starting from the Lagrangian, one can derive a Liouvillian that describes their time evolution as an additional term, e.g. $\hat{L}_V$.

Langevin-type thermostats [29] can also be described with an associated Liouvillian $\hat{L}_\xi$ and are a good example to explain the formalism. There are in fact multiple possible ways to factorize the overall Liouvillian: the so-called OBABO splitting reads $e^{\mathrm{i}\hat{L}_\xi \Delta t/2} e^{\mathrm{i}\hat{L}_p \Delta t/2} e^{\mathrm{i}\hat{L}_q \Delta t} e^{\mathrm{i}\hat{L}_q \Delta t/2} e^{\mathrm{i}\hat{L}_\xi \Delta t/2}$ and corresponds to bracketing a velocity Verlet integrator (BAB) between two finite-time propagators for an Ornstein-Uhlenbeck process (O, the Langevin equation for a free particle with inertia)

$$\begin{aligned}
\boldsymbol{p}_i &\leftarrow e^{-\gamma \Delta t/2} \boldsymbol{p}_i + \sqrt{m_i k_B T (1 - e^{-\gamma \Delta t})} \boldsymbol{\xi}_1 \\
\boldsymbol{p}_i &\leftarrow \boldsymbol{p}_i - \frac{1}{2}\frac{\partial V}{\partial \boldsymbol{q}_i}\Delta t \\
\boldsymbol{q}_i &\leftarrow \boldsymbol{q}_i + \frac{\boldsymbol{p}_i}{m_i}\Delta t \\
\boldsymbol{p}_i &\leftarrow \boldsymbol{p}_i - \frac{1}{2}\frac{\partial V}{\partial \boldsymbol{q}_i}\Delta t \\
\boldsymbol{p}_i &\leftarrow e^{-\gamma \Delta t/2} \boldsymbol{p}_i + \sqrt{m_i k_B T (1 - e^{-\gamma \Delta t})} \boldsymbol{\xi}_2
\end{aligned} \tag{9}$$

where $\boldsymbol{\xi}_1$ and $\boldsymbol{\xi}_2$ are vectors of uncorrelated, unit-variance Gaussian random numbers. Note that this splitting preserves the symmetry of the VV integrator. Many other splittings are possible, such as BAOAB (with the Ornstein-Uhlenbeck propagator sandwiched between two half-VV integrators), which has been found to be more accurate for position-dependent observables [91].

From this extensive overview of the Liouville operator formalism and its connection to the theory of integrators, one can see how to incorporate FlashMD into the integration schemes that are used for other thermodynamic ensembles. If the full Liouvillian for a given integrator is $\hat{L}' + \hat{L}_q + \hat{L}_p$, one can factor an integration over $\Delta\tau\Delta t$ as

$$e^{i\hat{L}'\Delta\tau\Delta t/2}(e^{i\hat{L}_p\Delta t/2}e^{i\hat{L}_q\Delta t}e^{i\hat{L}_q\Delta t/2})^{\Delta\tau}e^{i\hat{L}'\Delta\tau\Delta t/2}. \tag{10}$$

The central term is precisely the evolution that FlashMD aims to approximate, and can therefore be readily replaced with one large step on $(\boldsymbol{q}, \boldsymbol{p})$.

There are a few considerations that should be made when designing one of these extended integrators for FlashMD. First, if there are multiple splittings available for the base integrator, one has to choose those that involve a VV core. For instance OBABO can be used, but not BAOAB, and we cannot use the Bussi-Zykova-Parrinello splitting [92], but a more naive one that does not simultaneously update atomic positions and cell vectors. Second, if one wants to further split $e^{i\hat{L}'\Delta\tau\Delta t/2}$, it should be done in a symmetric way, applying the factors in opposite order before and after the FlashMD step. Last, and most importantly, the integration of $e^{i\hat{L}'\Delta\tau\Delta t/2}$ should be accurate also with a large time step, and the factorization with the VV core be similarly accurate, or at least preserve the target ensemble. This implies choosing large effective masses for extended Lagrangian terms (e.g. the cell volume in a constant-pressure integrator). Long time scales for Langevin-type thermostats should also be chosen if one wants to preserve time-dependent properties of the original Langevin dynamics – but this is a lesser concern for sampling accuracy, because usually Langevin-type free-particle integrators preserve the velocity distribution for any time step. The workflow proposed herein for the integration of thermostats and barostats with FlashMD is shown in Fig. 6.

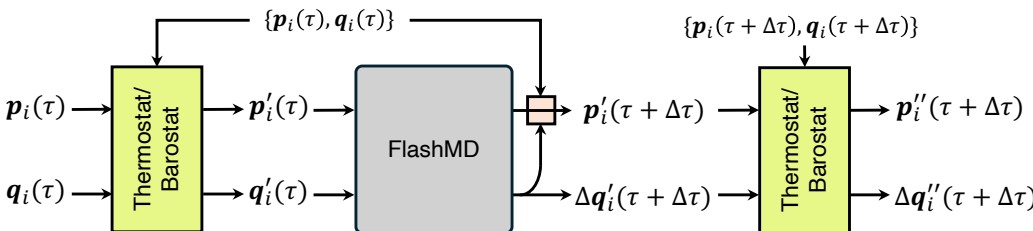

Figure 6: Integration of thermostats and barostats with FlashMD for thermodynamic ensemble control. Note that the integration is performed for the full stride in FlashMD, and the split operators on both ends are applied for half strides.

## E    Uncertainty quantification

ML models are inherently statistical, introducing different types of error at prediction time. In the context of FlashMD, uncertainty quantification (UQ) becomes an even bigger necessity as the potential error accumulation along a trajectory makes FlashMD simulations prone to many undesirable artifacts, leading to the incorrect sampling of the thermodynamic ensemble, which can manifest itself, for example, as unphysical bond forming/breaking behavior, uncontrolled expansion of the simulated system, etc. In this section, we identify the design choices that are required for robust UQ for MD prediction models and provide simple demonstrations.

In general, there largely exist two different sources of uncertainty for the model: *aleatoric*, irreducible uncertainty stemming from the "noise" in data, and *epistemic*, reducible uncertainty from the model's lack of knowledge [93]. In the training of MLIPs, it is widely assumed (although not necessarily true [94–96]) that the reference data is noise-free, and that it is therefore appropriate to only account for epistemic uncertainty in UQ approaches for MLIPs. In the learning formulation for FlashMD that directly targets time-evolved positions and momenta, we note the potentially significant presence of aleatoric uncertainty due to the chaoticity of the underlying physical problem. This aleatoric contribution to the uncertainty is expected to be strongly heteroscedastic, since different chemical systems exhibit very different degrees of chaotic behavior in molecular dynamics (in other words, the Lyapunov exponent can vary significantly based on the system, see Sec. 2.2 ). Last but not least, the UQ scheme of choice should not result in significant prediction time overhead in the simulation, as

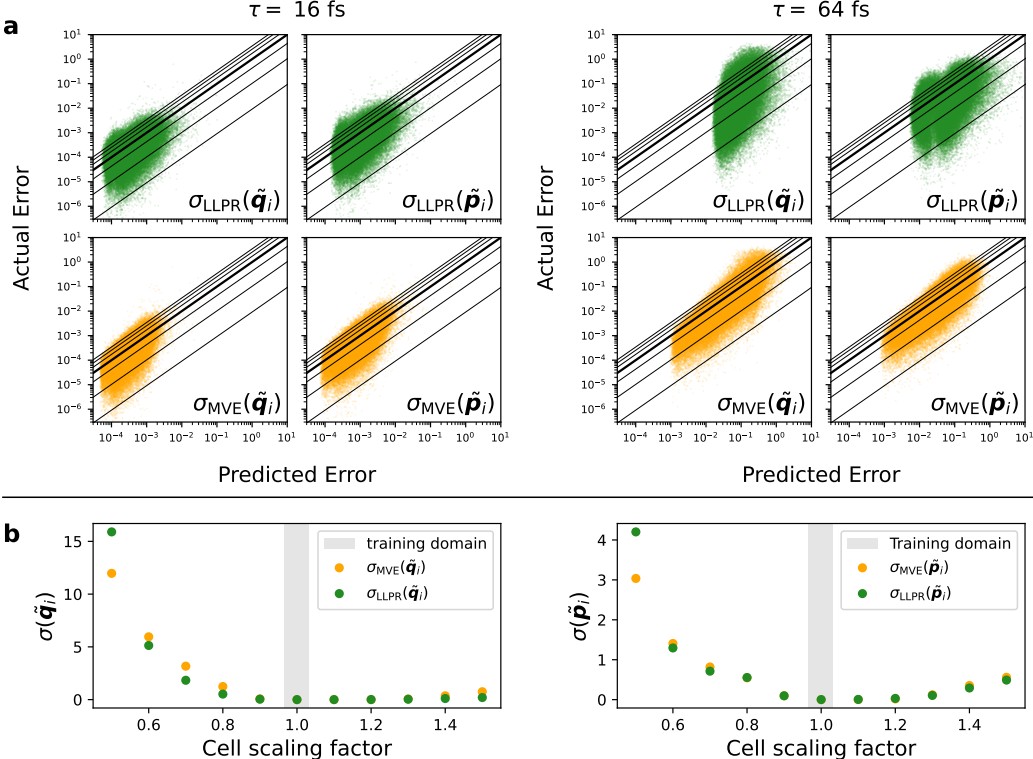

Figure 7: Uncertainty quantification diagnostics for the water-specific FlashMD model on its test set. (a) shows the uncertainty plots (predicted variance on the $x$ axis vs squared residual on the $y$ axis) on mass-scaled positions and momenta ($\tilde{q}_i$ and $\tilde{p}_i$) for two models trained on the water dataset (with 16 fs and 64 fs strides, respectively). Units are eV for ($\tilde{p}_i$) and Å$^2$u for ($\tilde{q}_i$). $\sigma_{\mathrm{LLPR}}$ is shown in green, and $\sigma_{\mathrm{MVE}}$ is shown in orange. The black lines corresponds to the parity line, as well as pairs of iso-probability lines of the ideal distribution containing density equivalent to that contained within $1\sigma$, $2\sigma$, $3\sigma$ of a Gaussian distribution. (b) shows the predicted uncertainty, in the same units, for out-of-distribution predictions using the 16 fs model, as a function of the scaling of the cell and the atoms within it.

this would undermine the key advantages of FlashMD. For these reasons, we adopt a UQ scheme that quantifies both types of uncertainties with a near-zero computational overhead. The method is sketched in the blue inset of Fig. 1.

Taking inspiration from Immer et al. [97], we assume the overall uncertainty to arise from a sum of an epistemic and an aleatoric term

$$\sigma^2 = \sigma_a^2 + \sigma_e^2, \tag{11}$$

which are predicted by different types of UQ estimator. For the aleatoric component, the prediction heads are modified to yield mean-variance estimators (MVEs), in which the model predicts mean and variance of the target predictions. FlashMD in this mode is trained to the negative log-likelihood loss

$$\mathcal{L} = \frac{1}{2}\left( \ln \sigma_a^2 + \frac{(y - y_{\mathrm{ref}})^2}{\sigma_a^2} \right), \tag{12}$$

where $\sigma_a^2$ is the variance afforded by the mean-variance estimator, parametrized as described in Lakshminarayanan et al. [98]. In our case, the overall loss is obtained by summing one of such terms for mass-scaled positions and one for mass-scaled momenta. A Laplace approximation is usually considered a good model for the epistemic uncertainty $\sigma_e^2$, and we implement it as a last-layer approximation (LLPR) [99–102] on the mean part of the MVE, i.e., on $y$.

We now demonstrate the UQ capabilities of FlashMD, with a special focus on the need for aleatoric uncertainty within direct MD prediction models as discussed in Sec. 3. To do so, we slightly deviate

from Eq. (11) and consider *separately* the MVE and LLPR uncertainty predictions for liquid water and analyze their behavior (Fig. 7a). The $\Delta\tau = 64$ fs model, which is very difficult to learn for the liquid water system (due to the fast correlation times of the physical system), displays the failure of epistemic uncertainty in isolation, exhibiting a narrow spread in values that does not provide meaningful insights. In contrast, the mean-variance estimator provides good uncertainty estimates. Perhaps more surprisingly, uncertainties of the 16 fs model are predicted reasonably by both uncertainty estimators. Since $\Delta\tau = 16$ fs is a more learnable regime for water, where epistemic uncertainty is expected to dominate, this implies that the mean-variance estimator is also capable of capturing epistemic uncertainties to good accuracy, at least within the training distribution. As an out-of-domain example, Fig. 7b shows the average LLPR and MVE uncertainties as a function of the compression (or expansion) factor of a water cell. Even for very compressed or very stretched cells (up to a change of 50% in the cell length), the mean-variance estimator provides a qualitatively correct uncertainty profile, suggesting that a MVE alone might be sufficient to quantify uncertainties in direct MD predictions. We leave a more thorough analysis of combining the two UQ metrics for future work.

## F  Ablation studies

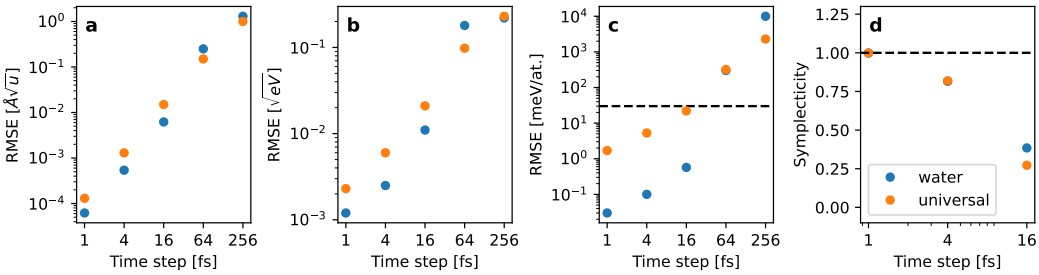

Figure 8: Comparison of validation root mean square errors (RMSEs) in (a) mass-scaled positions, (b) mass-scaled momenta, (c) energy conservation, and (d) symplecticity as a function of the stride for different FlashMD models. All errors are calculated on the respective validation sets, except for the symplecticity errors, which are evaluated using the left-hand side of Eq. 3 for 100 degrees of freedom of a liquid water structure. Dotted line in (c) is the RMSE of PET-MAD, an indirect metric of energy accuracy, and in (d) marks perfect symplecticity.

### F.1  Effect of the stride length on training

As a result of the chaoticity effects described in Sec. 2, it is desirable to examine the errors in the training as the predicted time stride increases. Fig. 8 shows the errors on energies and symplecticity, as well as mass-scaled positions and momenta, for the training runs on the water and universal datasets for different time steps. As expected, training becomes extremely difficult after a certain number of steps. From these plots, it would appear that the increase in the machine learning error is not exponential with the time step, but rather polynomial. While the interpretation of this observation is not trivial, it is potentially promising as it would make longer time scales accessible by simply improving the accuracy of the models using more data and/or learnable parameters, without the presence of hard limits to the accuracy. However, the rapid increase in the non-symplectic behavior of the predictions is alarming and worth future investigation.

### F.2  Enforcing energy conservation

We will now illustrate the effect of the procedure to enforce energy conservation presented in App. C on a simulation targeting the $NVE$ ensemble. Fig. 9 shows that, while FlashMD exhibits a total energy drift that would quickly lead to a unstable trajectories and large sampling errors, the proposed energy conservation enforcement technique allows for exact conservation of the total energy along the simulation, producing a stable trajectory (even thought it is not guaranteed to sample the $NVE$ ensemble because it is not exactly symplectic). We further show in Sec. 4 that this technique improves the temperature control in $NVT$ simulations.

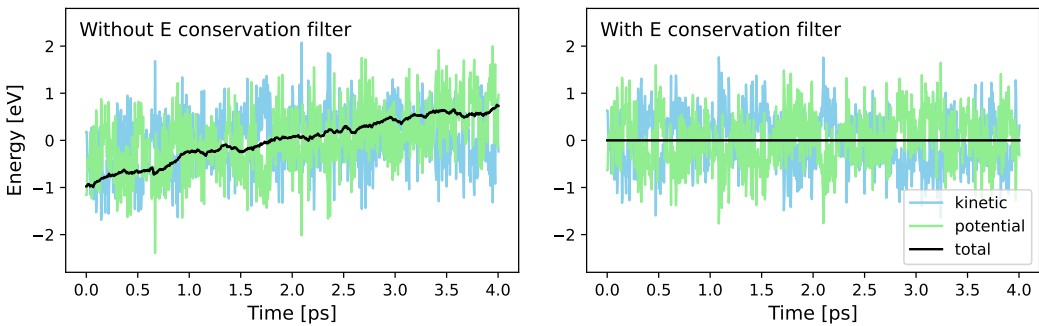

Figure 9: Comparison of the potential, kinetic, and total energies along short trajectories of water FlashMD model with 4 fs strides, with and without the energy conservation enforcement filter. The mean potential/kinetic/total energy is subtracted from all energy profiles.

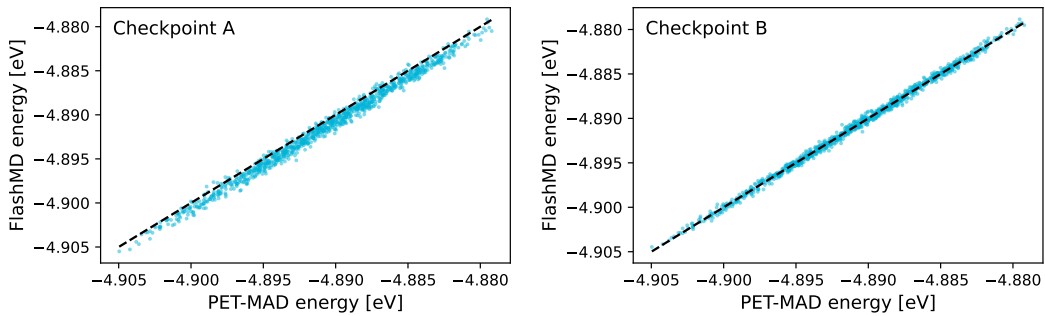

Figure 10: Parity plots of the predicted energy for two water-specific model checkpoints belonging to the same training run. The energies of the structure predicted by the model on the left are misaligned.

### F.3 Energy criterion for model selection

App. B describes how the error on the energy of the predicted structures is also used during training to choose good models. Here, to illustrate the utility of such approach, we show models from different epochs in the same training run (a water-specific model trained to predict over a time stride of 16 fs). Despite the two models having similar position and momentum errors, the energies of the first model are misaligned (Fig. 10). In a simulation, such an offset in model prediction would induce a progressive and unphysical cooling of the system.

## G   Timings

In this appendix, we report the overall timings for the simulations that were performed in this work. When multiple FlashMD models with different time steps were used, we report the largest-stride model that affords qualitatively accurate results. The timings obtained in this way are compiled in Table 2. In comparing these results with the theoretical speed-ups given by $\Delta\tau$, one should keep several issues into considerations: (i) We did not fine tune the time step of the base MD runs, nor attempt a fine-grained grid of acceleration factors for the FlashMD models. This might skew results in either directions: for instance, water can be run stably with 0.5 fs MD, and LiPS could have probably been run stably with a 32 fs FlashMD model. (ii) We did not systematically explore the Pareto frontier of FlashMD models in its architecture setup. However, in the current version, one evaluation of FlashMD is faster than a MLIP energy evaluation due to hyperparameter differences. (iii) Some of the extensions require additional energy evaluations; for instance energy scaling or the $NpT$ integrator require one or two energy evaluations each. There are obvious optimizations, such as only rescaling energy every few steps, which we did not consider in order to keep the analysis clearer for this first demonstration of a universal FlashMD model.

Table 2: Timings, strides, and acceleration factors of FlashMD compared to conventional MD on the systems investigated in this work. For completeness, we also include the speed-up factors we obtain without energy conservation enforcement (ECE), which would slightly degrade the quality of the simulations.

| System (# atoms) | Liquid water (192) | Al(110) slab (560) | Solvated alanine dipeptide (622) | Li$_3$PS$_4$ (768) |
|---|---|---|---|---|
| MD timing [stride] | $2.0 \cdot 10^4$ [0.25 fs] | $3.2 \cdot 10^4$ [1 fs] | $2.1 \cdot 10^5$ [0.5 fs] | $1.4 \cdot 10^5$ [2 fs] |
| FlashMD timing [stride] | $4.0 \cdot 10^2$ [16 fs] | $5.1 \cdot 10^2$ [64 fs] | $1.8 \cdot 10^4$ [16 fs] | $4.1 \cdot 10^4$ [16 fs] |
| Acceleration factor | **50** | **48** | **12*** | **3.4*** |
| Acceleration factor (no ECE) | **195** | **186** | **12*** | **3.4*** |

*These simulations are run in the $NpT$ ensemble, leading to more energy model evaluations in order to compute stresses. Re-using these energy evaluation for energy conservation enforcement and printing the energy to output, as opposed to recomputing them, would reduce the overhead, but it is not exploited in the present implementation. Furthermore, using non-conservative stresses [21] would speed up these simulations by a factor between two and three.

# H   Liquid argon benchmark (MDNet)

The early work by Zheng et al. [62] presents MDNet, a simple architecture for fitting molecular dynamics trajectories in the $NVE$ ensemble. Here, we compare FlashMD against MDNet, as well as Equivariant Graph Neural Networks [103] (EGNN, which was also explored as an alternative in Zheng et al. [62]). The lack of code and sufficient description makes it difficult to reproduce the workflow of the authors exactly; nonetheless, we attempt a similar set-up to Zheng et al. [62]:

- All reference simulations are run using LAMMPS, using a system of 256 argon atoms and a Lennard-Jones potential. The time step for the reference simulations is 1 fs.
- Ten runs are performed (eight for training, one for validation, one for testing), initially equilibrating in the $NpT$ ensemble for 100 ps, and then performing a production run for 10 ps in the $NVE$ ensemble using the velocity Verlet algorithm.
- From each $NVE$ trajectory, 25 equally spaced configurations are selected to be part of the training/validation/test set.
- A stride of 128 fs is used for training. Even though smaller strides are also explored in Zheng et al. [62], we find that this physical system is not particularly challenging due to its very simple and smooth potential energy surface, and therefore we only test predictions on the largest stride investigated in the original publication. Time-reversed targets are also added to the dataset, following Zheng et al. [62].

Using this set-up, we were able to reproduce the large-stride velocity Verlet results in Zheng et al. [62] exactly. The accuracies of velocity Verlet, EGNN, MDNet and FlashMD are shown in Table 3, where FlashMD is shown to outperform all methods. Note that our set-up involves less than one tenth of the training data used in Zheng et al. [62], and that FlashMD's accuracy is likely to be underestimated as a result.

Table 3: Accuracies of different methods for molecular dynamics trajectory predictions, on a liquid argon system with a predictive stride of 128 fs. EGNN and MDNet results are from Zheng et al. [62]. Positions errors are in units of Å, velocity errors are in units of Å/fs.

| Method | Velocity Verlet | EGNN | MDNet | FlashMD |
|---|---|---|---|---|
| RMSE ($q$) | $3.9 \cdot 10^{-2}$ | $1.3 \cdot 10^{-2}$ | $2.3 \cdot 10^{-3}$ | $\mathbf{5.4 \cdot 10^{-4}}$ |
| RMSE ($v$) | $1.8 \cdot 10^{-3}$ | $1.9 \cdot 10^{-4}$ | $5.7 \cdot 10^{-5}$ | $\mathbf{8.4 \cdot 10^{-6}}$ |

# I   SPC/E water benchmark (TrajCast)

TrajCast [63] published three datasets for the direct learning of molecular dynamics. Here, we compare our accuracies with the accuracies reported in Thiemann et al. [63], using the SPC/E water dataset they provide.

Table 4: Test-set accuracies of FlashMD and TrajCast [63] when trained on a SPC/E water dataset [63]. TrajCast results are reproduced from [63]. All errors are given in percentage MAE.

| Architecture | TrajCast | FlashMD |
|---|---|---|
| Displacements | 0.17 | 0.17 |
| Momenta | 0.37 | 0.22 |

The FlashMD results here make use of a slightly improved architecture compared to the one used to generate the results shown in this work. The same architecture was used to train the $r^2$SCAN FlashMD models we currently recommend, and it corresponds to the FlashMD implementation in `metatrain` [82]. The architecture used to produce the results in this work, for example, yields a momentum MAE of 0.52%, indicating that FlashMD and TrajCast have similar accuracies and that relatively minor tweaks can tip the numbers in favor of one or the other. In general, we always found our models to show clear signs of underfitting, showing that larger models and/or longer training times are generally beneficial when training models for the direct prediction of molecular dynamics trajectories, especially when compared to machine-learned interatomic potentials. The design of more accurate models will be a crucial challenge to achieve near-quantitative results using direct models for molecular dynamics in the future.

## J   Water simulations based on the q-TIP4P/f model

Due to the inaccurate description of water by PBEsol (the DFT functional used in the training of PET-MAD), we also train models on the q-TIP4P/f empirical water model [75] to investigate time-dependent properties in liquid water without raising the temperature, which in turns produces artifacts such as frequent bond dissociations that significantly affect the dynamics.

The dynamical properties we focus on are the mean square displacement (MSD) of oxygen atoms, as well as the dipole-dipole correlation function, both as a function of time. In order to avoid the large temperature deviations shown in Sec. 4 for the SVR thermostat, we instead use a fast-forward Langevin [104] thermostat, which is a modification of the Langevin thermostat aimed at reducing the effect of Langevin dynamics on dynamical properties, while being applied locally to each atom.

Fig. 11 shows the MSD and dipole-dipole correlation function for MD run with the q-TIP4P/f model, as well as FlashMD models of various strides. The results, shown in Fig. 11, establish the need for strong local thermostatting (time constant $\tau_L < 100$ fs) in order to obtain consistent statistical sampling between MD and FlashMD. Unfortunately, such thermostatting leads to an underestimation of the diffusion coefficient of water (which is proportional to the slope of the MSD curve) which is evident comparing the reference MD results for the $\tau_L = 100$ fs with the gentler $\tau_L = 1000$ fs. The tradeoff between thermostat strength and accuracy in enforcing correct sampling is similar to what was observed for explicit MD simulations based on non-conservative direct-force models [21]. This suggests that, as in that case, improving model accuracy and refining the thermostatting strategy might mitigate but not cure the underlying problems, and that one should also investigate methods to recover some of the conservation laws that are obeyed by MD trajectories, in order to achieve guaranteed quantitative accuracy in dynamics and sampling.

## K   Simulation details

All MD simulations in Sec. 4 were performed with `i-PI` [84], employing the internal implementation of the thermostats and barostats in the case of reference MD, and using custom, FlashMD-compatible integrators that adopt the integration schemes explained in Sec. D. In FlashMD simulations, inference was made with both the energy conservation enforcement filter and the random rotation filter, unless specified otherwise.

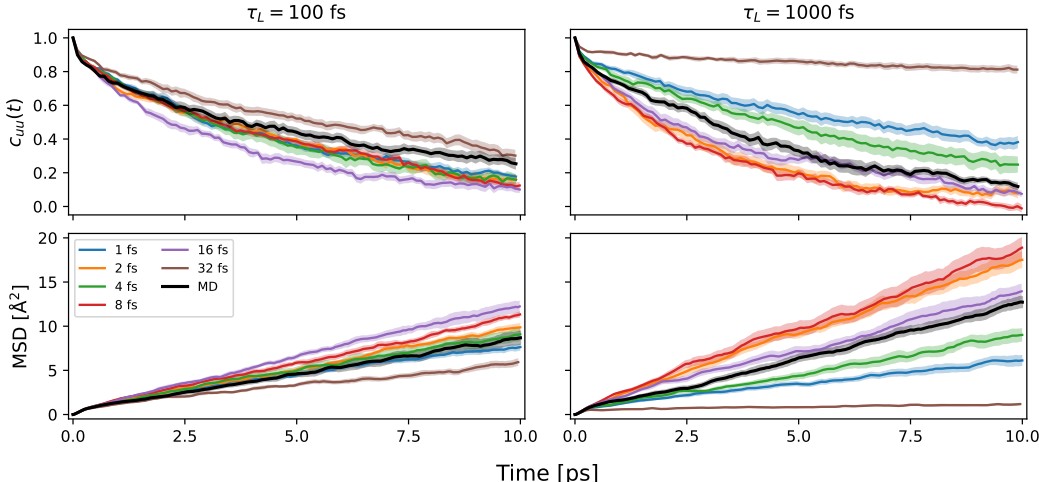

Figure 11: Dipole-dipole correlation functions (top) and mean-square displacement (bottom) for simulation of water using fast-forward Langevin thermostats with time constants $\tau_L = 100$ fs (left) and $\tau_L = 1000$ fs (right). The stronger thermostat leads to results from the different models that approach those of VV trajectories, but introduces its own spurious effects on dynamics.

**Water**    All PET-MAD-based water-specific model runs were performed for 100 ps using a periodic box of 64 water molecules and starting from a structure equilibrated with PET-MAD in the $NVT$ ensemble at 450 K, using a volume corresponding to the experimental density of liquid water at 300 K. $NpT$ trajectories were started from the same structure, using a temperature of 450 K and a pressure of 1 bar. The q-TIP4P/f-based models were equilibrated and run in the $NVT$ ensemble in the same way, but at 300 K and using the q-TIP4P/f for both equilibration and energy evaluations.

**Solvated alanine dipeptide**    $NpT$ simulations at 450 K and 1 bar were performed with a single dipeptide solvated by 200 molecules of water in a cubic cell of $20 \times 20 \times 20$ Å$^3$, as done in Morrone et al. [79]. The initial configuration of the system was randomly initialized with `packmol` [105], and the starting conformations of the dipeptide were taken from Ref. [106]. Simulations were performed for PET-MAD MLIP at 0.5 fs strides, universal FlashMD with 8 fs strides, and universal FlashMD with 16 fs strides, for a total duration of 1 ns. Langevin thermostat was coupled to the system with $\tau = 100$ fs, and an isotropic Bussi-Zykova-Parrinello (BZP) barostat [92] was used with $\tau = 400$ fs, including a Langevin thermostat coupled to the cell parameters with $\tau = 200$ fs. For each MD engine, 10 simulations were parallelly performed with different starting configurations to optimize sampling. Despite the difference in time strides, equivalent number of snapshots were sampled across the simulation setups to ensure a fair comparison of the resulting free energy surfaces.

**Al(110) surface**    Al(110) slab configurations were generated with ASE from a $5 \times 6 \times 8$ supercell and 20 Å vacuum in $z$ direction. Following Marzari et al. [76], $NVT$ simulations were performed for the system for the temperature range between 400 K and 900 K, for 0.5 ns. The system was coupled to a SVR thermostat [30] with $\tau = 10$ fs. PET-MAD MLIP at 1 fs strides, universal FlashMD at 16 fs strides, and universal FlashMD at 64 fs strides were used for the simulations.

$\gamma-$**Li$_3$PS$_4$**    Simulations details closely follow those of the original work by Gigli et al. [80] and the starting configuration was also obtained from the reference. 3 ns $NpT$ simulations at 0 bar were performed for temperatures between 575 and 725 K at 25 K intervals, using PET-MAD MLIP at 2 fs strides and universal FlashMD at 16 fs strides. SVR thermostat [30] was coupled to the system with $\tau = 10$ fs, and the BZP barostat [92] was used with $\tau = 1000$ fs, including a Generalized Langevin Equation [107] (GLE) thermostat coupled to the cell parameters at the same value of $\tau$. From the resulting MD trajectories, MSD of Li was first computed and used to calculate the Li conductivity with the Nernst-Einstein equation.

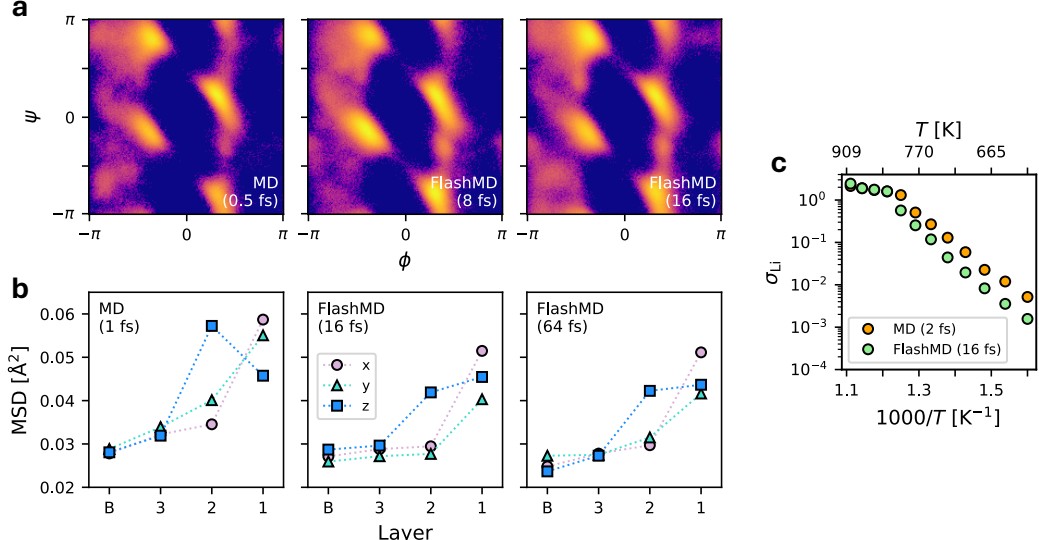

Figure 12: Results of case studies conducted for another set of universal FlashMD models trained with the trajectories from `PET-OMATPES`. (a) Ramachandran plots of the main backbone dihedrals for a simulation of solvated alanine dipeptide at 300 K. (b) Mean square displacement (MSD) of the Al (110) surface atoms at 500 K, at different layers from the surface (B indicates the limiting value for the bulk). (c) Li conductivities of $\gamma-Li_3PS_4$ at varying $T$.

## L    Univerals FlashMD models trained with `PET-OMATPES`

Here, we demonstrate the performance of another set of universal FlashMD models that adopt a different universal MLIP as the baseline. The new model of choice is `PET-OMATPES` [108], which is an MLIP that has first been trained on the OMat24 dataset [109], then fine-tuned on the MatPES-r2SCAN dataset [110]. Using this model, we followed the same protocol detailed in App. B to generate the reference trajectories that can be used for training, then trained the new set of universal FlashMD models. These models have also been made available in the FlashMD HuggingFace repository. The same set of case studies as the original universal FlashMD models (see Fig. 3) have been performed. We note that the alanine dipeptide case study was newly conducted at 300 K, as the r2SCAN functional does not suffer from the inaccuracies in describing aqueous systems that PBEsol functional has. The new set of results in Fig. 12 show that the `PET-OMATPES`-based universal FlashMD models successfully capture the expected trends in all three case studies.

## M    Chaos and Lyapunov exponents

Due to the chaotic nature of the Hamiltonian dynamics of the vast majority of non-trivial microscopic systems, small errors in the prediction of one step nearly always propagate to exponentially (in the elapsed time) large errors with respect to the reference trajectory. Due to this phenomenon, calculating errors in the reproduction of an individual trajectory is nearly meaningless, and what matters in practice is the correct description of collective dynamical modes. Therefore, instead of comparing the long-time accuracy of FlashMD against that of MD, a basic (but still meaningful) experiment is that of comparing the Lyapunov exponents [44] of the respective dynamics.

In the following table, we report the Lyapunov exponents of the dynamics of two different chemical systems: a liquid water system and a solid disordered GeTe system, both equilibrated at 300 K with standard MD with a Langevin thermostat and then run targeting NVE conditions both with MD and FlashMD, using the newer r$^2$SCAN models. We used 0.5 fs MD and 16 fs FlashMD for the water system, and 2 fs MD and 64 fs FlashMD for the GeTe system.

This result shows that FlashMD displays similar chaotic behavior to that of the baseline Hamiltonian dynamical system, although further evidence would be needed to establish these facts with certainty.

Table 5: Inverse of the largest Lyapunov exponents in the MD and FlashMD dynamics of water and GeTe.

| System | MD | FlashMD |
|--------|--------|---------|
| Water | 137 fs | 114 fs |
| GeTe | 294 fs | 238 fs |

## N  Computational resources

The use of computational resources in this work mainly stems from the generation of the universal dataset of $NVE$ trajectories, which employed 20,000 GPU hours on an Nvidia GH200 cluster. Model training was performed on Nvidia H100 GPUs, for a total of around 3,000 GPU hours. All other experiments, mostly molecular dynamics, were run either on H100 or L40S GPUs, and they do not contribute to the overall total compute in a significant way. Overall, we estimate our total usage of computational resources as slightly under 25000 H100 GPU hours.

