# OpenReview forum: "FlashMD: long-stride, universal prediction of molecular dynamics"
_NeurIPS.cc/2025/Conference — NeurIPS 2025 spotlight_

### Official Review · Reviewer_ZniE · 2025-06-28

**Clarity:** 4
**Significance:** 4
**Originality:** 2
**Rating:** 5
**Confidence:** 4

**Summary:**

This paper introduces FlashMD, a graph neural network-based framework for long-stride molecular dynamics (MD) prediction. Instead of integrating atomic trajectories with femtosecond-scale time steps, FlashMD directly predicts the positions and momenta of atoms at large time strides (up to 64 fs), bypassing both force evaluation and numerical integration.

The model is trained either on system-specific or universal MD trajectories generated using a machine-learned interatomic potential (PET-MAD), and it is designed to handle diverse thermodynamic ensembles (NVE, NVT, NpT). To stabilize predictions and maintain physical plausibility, the authors incorporate energy conservation at inference time, rotational and time-reversal augmentations during training, and uncertainty quantification to manage chaos and out-of-distribution behavior.

FlashMD is evaluated across a wide set of systems—including water, biomolecules, metals, and solid-state electrolytes—and shows significant speed-ups over conventional MD with competitive accuracy.

**Questions:**

1.	How does FlashMD compare to TrajCast or other equivariant autoregressive GNNs on realistic systems (e.g., alanine dipeptide)?
2.	How many steps can be chained in FlashMD before the trajectory diverges significantly from ground truth MD? Is there any cumulative error analysis?

**Ethical Concerns:**

["NO or VERY MINOR ethics concerns only"]

**Limitations:**

yes

**Quality:**

4

**Strengths And Weaknesses:**

Strengths
- FlashMD bypasses the standard time-integration paradigm by directly learning large time step updates in atomic trajectories—offering acceleration up to 34× in realistic settings.
- Universal generalization: The model is trained on a chemically diverse dataset (MAD) and evaluated across multiple systems and ensembles without fine-tuning, supporting the claim of universality.
- The model includes mass-scaled targets, center-of-mass conservation, optional rotational data augmentation, and energy conservation enforcement during inference.
- Strong empirical scope: Results span small molecules, biomolecules, and materials systems across different thermodynamic settings, with ablation studies and timing benchmarks.

Weaknesses
- The model is not symplectic by design. Although energy is conserved via momentum rescaling, phase space volume conservation and time reversibility are not guaranteed, raising questions about long-term stability and correctness.
- Key prior models such as TrajCast or Equivariant Graph Neural Operators are not evaluated head-to-head on realistic benchmarks. Comparisons to MDNet and EGNN are restricted to toy systems (liquid argon).
- Limited evidence for stability over long time horizons: The paper does not systematically assess long-time trajectory divergence, especially for chaotic systems. There is no quantitative analysis of error accumulation across multiple FlashMD steps.

---

> ### Author Rebuttal · Authors · 2025-07-29
>
> We thank the Reviewer for their comments, which are all very appropriate and relevant. Similar to the Reviewer, we also consider the lack of symplecticity by design to be the main limitation of this approach, and we hope that we will have more room to discuss it further in the camera-ready version of the manuscript, if it is accepted.
>
> Here are our answers to the questions:
> - We did not compare to TrajCast [1] explicitly because we had limited time to do so. TrajCast was released on arxiv six weeks before the submission deadline (incidentally, this means that it should be considered as “contemporaneous” work, according to the NeurIPS guidelines), and we came to know of it only three weeks after that. Nevertheless, we are happy to share some new results that we obtained during the rebuttal period with the Reviewer. We have trained FlashMD on the water trajectory dataset released by the authors of TrajCast (which is generated using the SPC force field as implemented in LAMMPS, and targets a 5 fs prediction step), and compared our results to the results that the TrajCast manuscript reports. FlashMD obtains out-of-the-box (i.e., without dataset-specific hyperparameter optimizations) a momentum MAE of 0.52%, while TrajCast achieves 0.37%. Although the results are comparable and both are accurate enough for stable direct MD propagation, it seems that TrajCast is more accurate on this particular dataset. Note that longer-time-step trajectory datasets were not made available by TrajCast authors, even though we were successfully able to push FlashMD water models for time strides up to 16 fs (i.e., more than three times as much as TrajCast). Still, from indirectly comparing the reported higher-time-step accuracies of TrajCast for 15 fs strides and FlashMD for 16 fs strides (trained on our own water trajectory dataset using the q-TIP4P/f force field), it is likely that our model performs better than TrajCast for longer strides. We would be happy to add these results and considerations in a revised version of our work. Finally, we would like to highlight that a direct MD prediction model is much more useful in practice if it is (1) pre-trained and applicable to the whole periodic table and (2) able to predict long strides to provide significant acceleration. These are two features that our models are designed for and manage to achieve in practice, while TrajCast might be, in our opinion, more limited in its scope.
>
> - We thank the Reviewer for this question and believe it is important to clarify the scope of the "long-term stability" issue that the Reviewer is referring to. In a statistical sense, our benchmarks demonstrate that FlashMD trajectories are very stable (they do not fail catastrophically) and in semi-quantitative agreement with the baseline short-time integrator (meaning that the ensemble they sample, and the average dynamical properties, are close to those of the reference MD). There are in all likelihood starting conditions or materials compositions that would lead to major instabilities (just as it would be the case for the parent MLIP), but we have not found any this far in our experiments, which involve a number of realistic and challenging systems and tens of millions of time steps. In the sense of “how long will it take until the actual trajectory will deviate completely from the specific MD trajectory” the answer is “a few steps, but it does not matter in practice”. MD is chaotic by nature and even with a VV integrator small perturbations of the initial configuration leads to exponential deviations in the trajectories (this can be modeled as $\Delta q \cdot \textrm{exp}(-t/\tau)$, where the time constant $\tau$ is often referred to as the Lyapunov exponent). This means that even a minuscule error in the predicted trajectory would lead to exponentially divergence of the trajectories, which however does not imply that one cannot extract accurate time-dependent properties from the simulation. We ran some additional experiments determining the Lyapunov exponents of MD and FlashMD. Both are very similar (for example, around 400 fs on a GeTe system) meaning that FlashMD is about as chaotic as the baseline. We intend to add these experiments in a revision of our work. These results are very interesting and we thank the Reviewer for prompting our investigation.
>
> [1] Thiemann, Fabian L., et al. "Force-free molecular dynamics through autoregressive equivariant networks." arXiv preprint arXiv:2503.23794 (2025).

---

> > ### Comment · Reviewer_ZniE · 2025-08-04
> > **Thank You for the Clarifications**
> >
> > Dear authors,
> >
> > thank you for your response.
> > Your answers are mostly what I had in mind and what I hoped for as answers.
> > Given my previous score, I'm happy to keep it.

---

### Official Review · Reviewer_5YjL · 2025-07-02

**Clarity:** 3
**Significance:** 2
**Originality:** 3
**Rating:** 5
**Confidence:** 3

**Summary:**

The paper introduces a deep learning model that predicts up to ~100 times larger time steps than MD simulation with machine learning force fields. The model is based on a GNN, which uses positions and velocities of the current state to predict a new configuration ~10-100 MD steps ahead. This potentially allows being faster than MD. The model is trained on a wide variety of systems, and can, hence, be evaluated even on unseen systems, which is demonstrated in the experiments. The investigated evaluation systems are water, alanine dipeptide in water, a metal surface, and a solid-state electrolyte, where the model recovers a phase transition.

**Questions:**

- There is a lot of work on ML models that predict larger time steps than MD, which are not discussed in this work. Although the here presented approach is slightly different, they should be cited and discussed regardless. Examples include "Timewarp: Transferable Acceleration of Molecular Dynamics by Learning Time-Coarsened Dynamics" by Klein et al. and "Implicit Transfer Operator Learning: Multiple Time-Resolution Surrogates for Molecular Dynamics" by Schreiner et al.
- The PET-MAD MLIP is used throughout the paper, I think it would help to have a short section discussing this model
-  The FlashMD framework section is a little short, a lot of important things, like the training, are only mentioned in the appendix. I am not sure that this can be fixed given the page limit, though.
- For the metal surface experiment: From figure 3c it seems that the transition temperature is predicted differently, while the text states that it matches. Why the discrepancy? Maybe there are some error bars that are not shown?
- I expect that the systems the model was evaluated on were not seen during training for the universal model. Is this correct? And how similar are the systems in the training set?

**Ethical Concerns:**

["NO or VERY MINOR ethics concerns only"]

**Final Justification:**

The authors addressed my questions and concerns in the rebuttal. Especially important was that the universal model was evaluated on systems unseen during training.

**Limitations:**

Yes

**Paper Formatting Concerns:**

No concerns

**Quality:**

3

**Strengths And Weaknesses:**

Strengths:
- The paper is well written and clearly motivated. Especially, MD simulations and potential pitfalls are in depth discussed.
- Although the idea of predicting larger time steps with ML models is not novel, the proposed architecture with the filtering is. Moreover, the model seems to be much more transferable than previous models.
- They evaluate the same trained model on a variety of systems, highlighting the transferability of the universal model
- The model can be applied to different ensembles, e.g.. NVE, NVT, NPT.

Weaknesses:
- For the alanine dipeptide experiments, the positive phi states are not shown.
- Results are mostly qualitative
- Wall-clock speed of the method is unclear. Although FlashMD works with larger time steps, it might still be slower than using an MLIP, e.g., PET-MAD. It most likely is slower than classical MD, but might be more accurate.
- See also questions

---

> ### Author Rebuttal · Authors · 2025-07-29
>
> We thank the Reviewer for their feedback and their questions. As the Reviewer pointed out, most molecular dynamics or sampling predictors are only valid for a single thermodynamic state point, corresponding to the trajectories they were trained on. By disentangling the stochasticity of thermostatted dynamics from the underlying (and purely deterministic) Hamiltonian dynamics, a single instance of our models is transferable to arbitrary thermodynamic conditions. From the responses of the other Reviewers, it is clear that this aspect was underemphasized in our initial manuscript, and we will work to make it clearer.
>
> Here are our answers to the questions:
> - We thank the Reviewer for this point. Such pre-existing models that seemingly deal with larger time strides adopt a generative and/or time-recurrent approach, and deviate from the deterministic and Markovian nature of MD. As such, we put a lot more emphasis on works closer to our “direct MD prediction” model training setup in Section 2.3 where we highlight previous works. Nevertheless, we inform the reviewer that our manuscript already highlights ITO by Schreiner et al. [1], and we are happy to newly acknowledge Timewarp by Klein et al. [2] and other related works in the camera-ready version.
>
> - The PET-MAD universal MLIP [3] was used in this work to generate a dataset of molecular dynamics trajectories for our universal model (although any other universal MLIP could have been used). The Reviewer is correct that a description of PET-MAD would be beneficial. We will include an overview of the model as an Appendix.
>
> - As the Reviewer correctly pointed out, many details on the FlashMD architecture and its training had to be relegated to the Appendix given the page limit. In case of acceptance, we will include a more detailed description in the camera-ready version, making use of the additional allowed page.
>
> - In the text for Figure 3(c), we state that the conductivities are “reasonably matched” with systematic under/over-estimations, not the predicted transition temperature (Tc). Also, Tc calculation can be sensitive to the simulation setup and the calculation details (e.g. compare our PET-MAD results with that in Mazitov et al. [3]). Despite the slight discrepancy of 25 K between FlashMD and our own reference MD results, we confirm that the Tc of FlashMD simulations falls within the range of reference values for PET-MAD.
>
> - The training set for the universal FlashMD models does not deliberately include any of the examples we considered. As a matter of fact, the 10,000 structures from the MAD dataset [4] that were used for MD simulations to generate the training set does not include any configurations of Li3PS4, elemental Al, or solvated alanine dipeptide, and only contain 1 random cluster, 1 slab, and 3 bulk configurations with the same elemental stoichiometry as water. We further highlight that the exercises presented in Figure 3 are purely extrapolative exercises for FlashMD.
>
> We would also like to touch on the weaknesses pointed out by the Reviewer, very briefly:
> - We only showed the negative-phi half of the Ramachandran plots as they are generally believed to be the most interesting part for solvated alanine dipeptide (e.g., see Fig. 4 of Morrone et al. [5], Fig. 3 of Koyama et al. [6]). During the rebuttal, however, we still confirmed that the simulations that we have used to generate the original plots show reasonable consistency also in the positive phi range. We would be happy to update our Figure 3a to also include the positive phi range in the camera-ready version of our manuscript.
>
> - Timings of the simulations are reported in Table 2 of Appendix G. FlashMD is consistently faster than MLIPs, although, as the Reviewer correctly points out, slower than MD with (less accurate and non-reactive) classical force fields.
>
> [1] Schreiner, Mathias, Ole Winther, and Simon Olsson. "Implicit transfer operator learning: Multiple time-resolution models for molecular dynamics." Advances in Neural Information Processing Systems 36 (2023): 36449-36462.
>
> [2] Klein, Leon, et al. "Timewarp: Transferable acceleration of molecular dynamics by learning time-coarsened dynamics." Advances in Neural Information Processing Systems 36 (2023): 52863-52883.
>
> [3] Mazitov, Arslan, et al. "PET-MAD, a universal interatomic potential for advanced materials modeling." arXiv preprint arXiv:2503.14118 (2025).
>
> [4] Mazitov, Arslan, et al. "Massive Atomic Diversity: a compact universal dataset for atomistic machine learning." arXiv preprint arXiv:2506.19674 (2025).
>
> [5] Morrone, Joseph A., et al. "Efficient multiple time scale molecular dynamics: Using colored noise thermostats to stabilize resonances." The Journal of chemical physics 134.1 (2011).
>
> [6] Koyama, Yohei M., Tetsuya J. Kobayashi, and Hiroki R. Ueda. "Perturbation analyses of intermolecular interactions." Physical Review E—Statistical, Nonlinear, and Soft Matter Physics 84.2 (2011): 026704.

---

> > ### Comment · Reviewer_5YjL · 2025-08-05
> >
> > I thank the authors for their detailed rebuttal, all my concerns have been addressed. I increased my score accordingly.

---

> > > ### Author Response · Authors · 2025-08-05
> > >
> > > We appreciate the Reviewer’s feedback and the time they dedicated to evaluating our manuscript.

---

### Official Review · Reviewer_SKE5 · 2025-07-02

**Clarity:** 3
**Significance:** 3
**Originality:** 3
**Rating:** 5
**Confidence:** 3

**Summary:**

This manuscript investigates the problem of learning a “universal” model for directly predicting state changes (coordinates and momenta) as an alternative approach to molecular dynamics (MD) simulations. Unlike previous methods such as MLIPs that focus on energy/force prediction or direct MD propagators on specific systems, the proposed approach aims for broader applicability and faster acceleration. The authors discuss several critical factors for training such a propagator and propose solutions, including the choice of GNN architecture, data augmentation strategies, and energy conservation during both training and inference. In experiments, they demonstrate that the proposed FlashMD model can accurately recover MD ensemble statistics for both single-system cases (e.g., water) and across diverse chemical systems.

**Questions:**

1. Have the authors attempted to train a single model across multiple temporal scales? What are the potential challenges in learning such multi-timescale dynamics within one model?

1. Can FlashMD be trained on MD datasets that do not follow the NVE ensemble or lack full observations (e.g., missing momenta, solvent state not saved)? How critical are the assumptions of determinism and strict Markovian dynamics in such cases?

1. The authors evaluate energy conservation error for model selection but do not include it as part of the training loss. What are the main challenges in incorporating energy loss directly during training?

1. All experiments are compared only against reference MD from PET-MAD. Why were no machine learning baselines, particularly other “direct MD propagators” discussed in Section 2.3, included for comparison?

1. Scalability and generalization: Current evaluations seem to be limited to small systems (<1000 particles) and short simulations (ps-level). Have the authors tested the model on larger systems or longer timescales?

**Ethical Concerns:**

["NO or VERY MINOR ethics concerns only"]

**Final Justification:**

The authors has addressed my main questions (multi-timescale, training on NVE, energy conservation, prediction strides). Remaining issue is minor and universal (scalability). The manuscript is pleasant to read and insightful. Recommend for acceptance.

**Limitations:**

Yes.

**Paper Formatting Concerns:**

No concerns.

**Quality:**

4

**Strengths And Weaknesses:**

## Strengths

1. The manuscript is clearly written and well structured. The main objectives are effectively conveyed, and the authors provide a comprehensive review of the problem, in-depth discussions, and detailed results in the appendix. Overall, the clarity and  quality are high.
2. The paper offers a solid review of the physical and mathematical foundations of molecular dynamics, demonstrating a strong grounding for the proposed method.
3. The ablation studies are thorough, covering key factors such as training time steps and enforcement of energy conservation. Showing the importance of considering these key factors.
4. Results show that a universal, direct MD propagator model can potentially capture the underlying physics across different systems.

## Weaknesses

1. A key limitation is the need to train separate models for different time strides, along with tedious hyperparameter tuning to find an appropriate stride that balances accuracy and speed. Especially for the “universal model,” optimal strides may vary across systems, this limits the model’s “universality” and practical applicability (significance). (Ref. Q1)

1. Some modeling assumptions, though physically sound, may be overly restrictive. For instance, the insistence on determinism and strict Markovian behavior limits the model learn from NVE simulation data with complete observability. In practice, many MD datasets include missing data (e.g., coordinates only on non-solvent atoms) or follow alternative ensembles (e.g., NPT), which violates these assumptions. (Ref. Q2)

1. The model is mainly evaluated against reference MD, with no comparison to other “direct MD propagator models”. This makes it difficult to assess the actual improvement of the proposed approach (Q4). Additionally, evaluations are conducted on small systems with short simulation times (Q5), limiting the scope of validation. (significance)

---

> ### Author Rebuttal · Authors · 2025-07-29
>
> We thank the Reviewer for their time and their feedback. On our side, it was pleasing to receive recognition for our efforts to formally ground our work in the fundamentals of molecular dynamics, which are often overlooked in similar works. Below are our responses to the Reviewer’s questions:
>
> 1. What the Reviewer refers to would constitute a very natural extension of our work, and it is a very relevant question. We have indeed tried to train a single model across multiple temporal scales. We found that the learning exercise becomes very difficult due to the need to learn the larger steps, where potentially many systems in the training set are in a chaotic (and therefore unlearnable, with current techniques) regime. Unfortunately, this negatively impacts the quality of the predictions also on smaller time steps. One additional consideration that made us lean towards training a single model per time step is that of computational efficiency. Learning arbitrary time horizons with a single model might require larger (and therefore slower) models, in a research field where simulation time is often a limiting factor. In this case, we believe that training and delivering one model per time step is more advantageous, especially from the point of view of practitioners. We thank the Reviewer for this question and we would be happy to include a discussion along these lines in a revised version.
>
> 2. We would like to begin by emphasizing that, from the point of view of the target ensembles, training on NVE data is not a limitation, but a deliberate choice that makes our model more transferable: we demonstrate the use of FlashMD in NVT and NPT sampling by combining the NVE “core” with completely inexpensive thermostat steps, and with a barostat that can be implemented with a couple MLIP evaluations. In other words, compared to generative methods, our approach disentangles Hamiltonian dynamics (which we learn exclusively from NVE data through a GNN) from, e.g., thermostatting, which can be applied separately and in a very inexpensive fashion. This allows us to generalize a single model to make predictions for arbitrary thermodynamic ensembles and arbitrary thermodynamic state points, which is something that generative models targeting specific distributions (i.e., a specific ensemble at a single thermodynamic state point) cannot achieve. That said, we see no major hurdle including the cell degrees of freedom to train on NPH trajectories to enable constant-enthalpy/constant-pressure runs without the additional MLIP evaluations that are included in our barostat. Removing the requirement of determinism and complete information from the training data should be possible (similar to how coarse-grained models are trained using the same functional form of a MLIP, but using only a subset of the degrees of freedom), but we have not tried it yet. It is certainly an interesting future direction, as it would require including an intrinsic stochastic component in the model architecture.
>
> 3. We thank the Reviewer for this very good question. We have tried including an energy conservation term in the loss, but we found no significant improvement in the energy conservation of the resulting trajectories, which is instead dominated by highly non-trivial error accumulation phenomena over tens or thousands of steps. We found that long-time energy conservation can only be enforced reliably by having an exactly symplectic model, and this is in complete agreement with the literature on numerical integrators for Hamiltonian systems [1]. (We experimented with symplectic architectures, but we found them to be too slow for molecular dynamics applications. This is a very promising future direction and we will definitely include a discussion of this aspect in a future version with less strict page limits.) Given that encouraging energy conservation through the loss function has only very limited positive effects on the quality of the resulting dynamics, and that evaluating an MLIP at every training step makes the training procedure nearly twice as expensive, we decided to train without this feature. If the Reviewer thinks this is helpful, we can definitely add a few sentences to explain why we decided not to add an energy conservation term in the loss for our models.
>
> 4. As the Reviewer points out, we identified two methods that are directly comparable to ours: MDNet and TrajCast. MDNet was presented in a very short, proof-of-principle, workshop paper [2] which only evaluated on what could be considered a "toy system": that of liquid argon with a Lennard-Jones potential. Since the MDNet code is not available (as far as we know), we had to use the instructions in the original paper to compare with the published results. This is what we did in Appendix H. We did not believe that this Lennard-Jones example was significant enough to appear in the main text, especially given the page limit. Regarding TrajCast [3], we did not compare to TrajCast explicitly because we had limited time to do so. TrajCast was released on arxiv six weeks before the submission deadline (incidentally, this means that it should be considered as “contemporaneous” work, according to the NeurIPS guidelines), and we came to know of it only three weeks after that. Nevertheless, we are happy to share some new results that we obtained during the rebuttal period with the Reviewer. We have trained FlashMD on the water trajectory dataset released by the authors of TrajCast (which is generated using the SPC force field as implemented in LAMMPS, and targets a 5 fs prediction step), and compared our results to the results that the TrajCast manuscript reports. FlashMD obtains out-of-the-box (i.e., without dataset-specific hyperparameter optimizations) a momentum MAE of 0.52%, while TrajCast achieves 0.37%. Although the results are comparable and both are accurate enough for stable direct MD propagation, it seems that TrajCast is more accurate on this particular dataset. Note that longer-time-step trajectory datasets were not made available by TrajCast authors, even though we were successfully able to push FlashMD water models for time strides up to 16 fs (i.e., more than three times as much as TrajCast). Still, from indirectly comparing the reported higher-time-step accuracies of TrajCast for 15 fs strides and FlashMD for 16 fs strides (trained on our own water trajectory dataset using the q-TIP4P/f force field), it is likely that our model performs better than TrajCast for longer strides. We would be happy to add these results and considerations in a revised version of our work. Finally, we would like to highlight that a direct MD prediction model is much more useful in practice if it is (1) pre-trained and applicable to the whole periodic table and (2) able to predict long strides to provide significant acceleration. These are two features that our models are designed for and manage to achieve in practice, while TrajCast might be, in our opinion, more limited in its scope.
>
> 5. Our framework has roughly the same scalability as current universal MLIPs. The computational effort scales linearly with system size, although it is limited to a few tens of thousands atoms on a single GPU due to memory reasons. We would be happy to add some scaling plots to the appendices if this Reviewer thinks it would be useful.  Current universal MLIPs can be applied to large-scale simulations by using multiple (10-100) GPUs in parallel [4], which has allowed machine-learning models to simulate up to around ten million atoms. We are currently working on the integration of our framework with LAMMPS, which involves a few technical hurdles, but no conceptual problem. Once this integration is complete, we expect to achieve a speed on the order of 10 ns per day, or slightly more, on such large systems. However, we would like to emphasize that, while these scales are sometimes needed to perform biochemical simulations, most research on materials using molecular dynamics does not need these scaling efforts and is performed on much smaller systems (between 500 and 5000 atoms).
>
> [1] Hairer, Ernst, et al. "Geometric numerical integration." Oberwolfach Reports 3.1 (2006): 805-882.
>
> [2] Zheng, Tianze, Weihao Gao, and Chong Wang. "Learning large-time-step molecular dynamics with graph neural networks." arXiv preprint arXiv:2111.15176 (2021).
>
> [3] Thiemann, Fabian L., et al. "Force-free molecular dynamics through autoregressive equivariant networks." arXiv preprint arXiv:2503.23794 (2025).
>
> [4] Musaelian, Albert, et al. "Learning local equivariant representations for large-scale atomistic dynamics." Nature Communications 14.1 (2023): 579.

---

> ### Comment · Reviewer_SKE5 · 2025-08-04
>
> I thank the authors for their thorough responses. I believe this work makes valuable contributions to the general MLFF field, and I am happy to adjust my scores accordingly. That said, a few limitations remain:
>
> 1. It seems that TrajCast and FlashMD have comparable results in head-to-head comparison and the main strengths of FlashMD is longer strides and broader coverage over element types.
> 2. The model's performance on larger systems remains unverified, while feasible with additional engineering efforts. It is a stretch and not a major concern.
>
> Additional comments:
>
> - On Q3: Yes, as a reader, I found the discussion around energy conservation and model trade-offs to be particularly important. I recommend briefly discussing this in the main text (methods or limitations) with further details in the appendix.
>
> - On Q4: Yes, please include a comparison with TrajCast in the final version.
>
> - On Q5: Scaling plots would be helpful in the final version.

---

> > ### Author Response · Authors · 2025-08-05
> >
> > We thank the Reviewer for their response and we agree with their additional comments and assessment.
> >
> > With regards to their point 1, we would briefly like to stress that achieving stable dynamics with longer time strides directly translates into greater efficiency in simulations, and that zero-shot transferability across thermodynamic ensembles and chemical compositions is a highly non-trivial and technically challenging advancement that FlashMD has demonstrated. For context, it took 15 years for machine-learning interatomic potentials (MLIPs) to progress from single-purpose models to universal/foundation models.
> >
> > We will work in the changes that have been discussed in Q3, Q4 and Q5 (which is related to point 2) in the camera-ready version of our manuscript, if it is accepted. We thank the Reviewer for suggesting these improvements to our work.

---

### Official Review · Reviewer_7QyG · 2025-07-03

**Clarity:** 2
**Significance:** 3
**Originality:** 3
**Rating:** 4
**Confidence:** 3

**Summary:**

This paper introduces FlashMD, a machine learning framework designed to accelerate Molecular Dynamics (MD) simulations. Instead of calculating forces at every minuscule time step, FlashMD uses a Graph Neural Network (GNN) to directly predict the positions and momenta of atoms over time strides that are up to two orders of magnitude longer than those used in traditional methods.

**Questions:**

(1) What is the total simulation time of the result in Figure 3(a) shows? \
(2) The experiments typically use hundreds of atoms - however, for practical purposes, it is common to use millions of atoms. What is the scalability of this framework?

**Ethical Concerns:**

["NO or VERY MINOR ethics concerns only"]

**Final Justification:**

The quality of the results is clarified by the authors, thus raising the score.

**Limitations:**

yes

**Quality:**

3

**Strengths And Weaknesses:**

**Strengths:**\
(1) Traditional MD approaches are typically known for their requirement of fine-grained time steps, leading to slow simulations. Approaches toward increasing strides should be encouraged. \
(2) The connection between MD and GNN is natural and worth investigating. \
**Weaknesses:**\
(1) Imprecise statements. \
1a) Line 55 is vague and can be misleading. The Hamilton's equations require energy conservation and are improper to directly define MD. Although the authors emphasize "In its simplest form" to make it more precise, this can still be misleading for researchers outside the research domain.
1b) VV approximately conserves energy only because it is applied to an energy-conserved system -- VV can be used without energy conservation, and it is misleading to only emphasize that VV is suitable for NVE due to the ability to conserve energy with sufficiently small time steps. \
1c) MD is not necessarily deterministic. e.g., some simulations take Brownian motion into consideration, and some include quantum effects. \
(2) Unclear central point. The paper should more focus on the key contribution, which I presume is the orders-of-magnitude increase in time step sizes. However, in the framework section, I only see scattered bullet points and fail to observe their connections. I would recommend focusing on a key point within the 9 pages or submit elsewhere with higher page limits. At current stage, the main text of this paper looks like a list of contents in appendix.

---

> ### Author Rebuttal · Authors · 2025-07-29
>
> We thank the Reviewer for their time and effort. We are happy to read that the Reviewer has identified the push towards larger time strides in MD simulations using GNNs as a worthwhile initiative and hence a strength of our work. Regarding the weaknesses that the Reviewer has pointed out, we would like to first make the following clarifications to help resolve their concerns about our work:
>
> 1a) The introduction we provide on the subject of molecular dynamics is in agreement with standard textbooks [1, 2] and graduate courses. When stating that our presentation is imprecise, we believe that the Reviewer refers to most practical applications of molecular dynamics, where thermostats are added to the time-integration algorithm, causing fluctuations in the energy which are consistent with the target thermodynamic ensemble. (We are not entirely sure that this is what the Reviewer was referring to and we invite them to inform us through a response if this was not the case.) We would like to emphasize that one of the advantages of our approach, compared to generative methods, is to disentangle Hamiltonian dynamics (which we learn exclusively from NVE data through a GNN) from thermostatting, which can be applied separately and in a very inexpensive fashion. This allows us to generalize a single model to make predictions for arbitrary thermodynamic ensembles and arbitrary thermodynamic state points, which is something that generative models targeting specific distributions (i.e., a specific ensemble at a single thermodynamic state point) cannot achieve.
>
> 1b) Here, we believe that the Reviewer is again referring to thermostatted systems, or perhaps systems with external forces, and we would like them to correct us if this is not the case. We clearly state the assumptions that lead to our treatment (e.g., “in the absence of external perturbations”, line 57), and therefore we believe that it is incorrect to characterize our introduction as “imprecise” or “vague”. If the Reviewer is more familiar with other sets of assumptions, we would be happy to make a comparison or to mention them as a limitation of our method. For instance, it is true that velocity Verlet can be applied to non-Hamiltonian systems (or non-separable Hamiltonian systems), and that the symplecticity of the algorithm, and therefore its long-time energy conservation [3], does not apply in those settings.
>
> 1c) We refer to 1a) for Langevin dynamics (which is equivalent to Brownian motion and can be seen as a form of thermostatting). Concerning nuclear quantum effects, path integral molecular dynamics can also be reduced to Hamiltonian dynamics and learned with FlashMD. We would be happy to discuss this point more in detail, both with the Reviewer and in the text of the manuscript.
>
> (2) We thank the Reviewer for pointing this out and we agree. Just for completeness, the central contributions are:
> (a) Speeding up molecular dynamics with large time steps; (b) Compared to generative approaches in the same domain, we learn from NVE data to generalize to arbitrary thermodynamic conditions with a single model; (c) Training a universal model (in the chemical sense, i.e., across the whole periodic table) for molecular dynamics predictions. We will definitely clarify and emphasize these points over more technical details in a revised version.
>
> Here are our responses to the Reviewer’s questions:
>
> (1) The total simulation time of the results in Figure 3(a) is 0.5 ns for each of the 10 runs, for all models. We thank the Reviewer for highlighting that this was missing. We will update the simulation details accordingly. We would like to emphasize that we chose these examples because they could be brought close to statistical convergence with a limited effort (approximately one GPU day per trajectory) even when performing the baseline MD simulations with an MLIP. As apparent from the timings in Table 2, the FlashMD models would allow for much longer simulation times with the same effort.
>
> (2) Our framework has the same scalability as current MLIPs. The computational effort scales linearly with system size, although it is limited to a few tens of thousands atoms on a single GPU due to memory reasons. We would be happy to add some scaling plots to the appendices if this Reviewer thinks it would be useful.  Current MLIPs can be applied to large-scale simulations by using multiple (10-100) GPUs in parallel [4], which has allowed machine-learning models to simulate up to around ten million atoms. We are currently working on the integration of our framework with LAMMPS, which involves a few technical hurdles, but no conceptual problem. Once this integration is complete, we expect to achieve a speed on the order of 10 ns per day, or slightly more, on such large systems. However, we would like to emphasize that, while these scales are sometimes needed to perform biochemical simulations, most research on materials using molecular dynamics does not need these scaling efforts and is performed on much smaller systems (between 500 and 5000 atoms).
>
> [1] Frenkel, Daan, and Berend Smit. Understanding molecular simulation: from algorithms to applications. Elsevier, 2023.
>
> [2] Allen, Michael P., and Dominic J. Tildesley. Computer simulation of liquids. Oxford university press, 2017.
>
> [3] Hairer, Ernst, et al. "Geometric numerical integration." Oberwolfach Reports 3.1 (2006): 805-882.
>
> [4] Musaelian, Albert, et al. "Learning local equivariant representations for large-scale atomistic dynamics." Nature Communications 14.1 (2023): 579.

---

> > ### Comment · Reviewer_7QyG · 2025-08-05
> >
> > I sincerely thank the authors for their detailed response and clarifications. I indeed referred to MD in a more general sense and recognized the scope of this paper (Sec. 2.1) as a general definition of MD. Still, I suggest defining the paper's scope early, as it is not straightforward enough to realize that the paper's scope is within a subset (NVE ensemble) of MD until Sec. 2.1.
> >
> > The authors' response with respect to the experimental results is an important factor - 0.5 ns is relatively long compared to the strides, and the figure thus shows good results without notable divergence. I would appreciate it if the authors can also discuss for how long the approach will break.

---

> > > ### Author Response · Authors · 2025-08-05
> > >
> > > We thank the Reviewer for their prompt reply and for their additional comments.
> > >
> > > We are glad we could clarify the scope of our manuscript. When revising, we will explain clearly in an earlier part of the manuscript that we target NVE for training because it allows us to perform nearly all relevant types of MD (including NVT, NPT, atom exchanges, etc., and at arbitrary thermodynamic state points) with a single model in production. In fact, some kind of weak thermostatting is needed to compensate for lack of energy conservation, and this stabilizes the trajectories indefinitely. To reply to the last question raised by the Reviewer, we ran tens of millions of time steps for a number of very diverse systems, and we never observed any catastrophic failures.
> > >
> > > We thank the Reviewer for their engagement and we remain available for any additional questions.

---

### Note · Authors · 2025-08-12

Dear Reviewers, Dear AC,

Thank you for your consideration. We are glad that the importance and soundness of our contribution was appreciated. We believe that our method will enable researchers to accelerate molecular dynamics significantly, while retaining the qualitative physical and chemical properties of the simulated system. We think that the possibility of sampling arbitrary thermodynamic ensembles, as well as the demonstration of a pre-trained universal models that can simulate diverse systems across the periodic table, will make our contribution especially useful for the community.

We appreciate your time and your valuable suggestions, which we are already working to incorporate into the manuscript.

---

### Decision · Program_Chairs · 2025-09-17

**Decision:**

Accept (spotlight)

**Comment:**

This paper presents an interesting new approach to enable larger time-steps in molecular simulations. The paper is well written, and clearly presents what task it is solving. The strengths of the approach is that is transferable. While the approaches not compete with generative molecular dynamics emulators in terms of time-steps possible, e.g. implicit transfer operators and timewarp, it provides and interesting general purpose approach to the problem, which will likely have an immediate impact on the molecular simulation field. Simultaneously, the approach can likely inform and inspire other approaches in related areas. All the weaknesses, raised by reviewers were thoroughly addressed addressed during the rebuttal. We hope the authors, include a thorough discussion of other related approaches in the camera ready version of this manuscript. This discussion should, include methods, such as those mentioned above, and more clearly position their own work in relation to these other approaches, which are orthogonal, rather than competing.